# A modular approach for modeling the cell cycle based on functional response curves

**Jolan De Boeck**[1,2], **Jan Rombouts**[1], **Lendert Gelens**[1]*

**1** Laboratory of Dynamics in Biological Systems, Department of Cellular and Molecular Medicine, University of Leuven, Leuven, Belgium, **2** Stem Cell Institute Leuven, Department of Development and Regeneration, University of Leuven, Leuven, Belgium

* lendert.gelens@kuleuven.be

**Data Availability Statement:** The code used to generate the computational results can be found at https://github.com/Gelens-Lab/cellcyclemodules.

## Abstract

Modeling biochemical reactions by means of differential equations often results in systems with a large number of variables and parameters. As this might complicate the interpretation and generalization of the obtained results, it is often desirable to reduce the complexity of the model. One way to accomplish this is by replacing the detailed reaction mechanisms of certain modules in the model by a mathematical expression that qualitatively describes the dynamical behavior of these modules. Such an approach has been widely adopted for ultrasensitive responses, for which underlying reaction mechanisms are often replaced by a single Hill function. Also time delays are usually accounted for by using an explicit delay in delay differential equations. In contrast, however, S-shaped response curves, which by definition have multiple output values for certain input values and are often encountered in bistable systems, are not easily modeled in such an explicit way. Here, we extend the classical Hill function into a mathematical expression that can be used to describe both ultrasensitive and S-shaped responses. We show how three ubiquitous modules (ultrasensitive responses, S-shaped responses and time delays) can be combined in different configurations and explore the dynamics of these systems. As an example, we apply our strategy to set up a model of the cell cycle consisting of multiple bistable switches, which can incorporate events such as DNA damage and coupling to the circadian clock in a phenomenological way.

## Author summary

Bistability plays an important role in many biochemical processes and typically emerges from complex interaction patterns such as positive and double negative feedback loops. Here, we propose to theoretically study the effect of bistability in a larger interaction network. We explicitly incorporate a functional expression describing an S-shaped input-output curve in the model equations, without the need for considering the underlying biochemical events. This expression can be converted into a functional module for an ultrasensitive response, and a time delay is easily included as well. Exploiting the fact that several of these modules can easily be combined in larger networks, we construct a cell

**Funding:** This work was supported by the Research Foundation Flanders (FWO, www.fwo.be) with individual support to J.D.B. (1189120N) and J. R. (11D0920N) and project support to L.G. (Grant GOA5317N) and the KU Leuven Research Fund (No. C14/18/084) to L.G. The funders had no role in study design, data collection and analysis, decision to publish, or preparation of the manuscript.

**Competing interests:** The authors have declared that no competing interests exist.

cycle model consisting of multiple bistable switches and show how this approach can account for a number of known properties of the cell cycle.

## Introduction

Cell division and the correct separation of genomic material among daughter cells is fundamental for the proper development, growth and reproduction of a living organism. The molecular mechanisms that underlie these processes are highly evolutionarily conserved. Incorrect cell division can have detrimental effects, ranging from developmental defects to the transformation of healthy somatic cells into cancer cells. Because of this, tight regulatory mechanisms are established early in embryonic development to ensure the correct replication of DNA and cell division. These control mechanisms, or cellular checkpoints, ensure that the cell cycle only progresses to its next phase if appropriate intra- and extracellular conditions are fulfilled. These conditions include the absence of DNA damage, proper alignment of the chromosomes in the metaphase plane, and abundance of nutrients and growth factors [1]. The sequential nature of cell cycle progression where the start of one phase depends on the completion of a previous phase resulted in the view of the cell cycle as a 'domino-like' process [2, 3].

Over the years, both theoretical and experimental studies have demonstrated how the mechanisms that control the 'domino-like' nature of cell cycle progression are centered around bistable switches. Such bistability is often encountered in systems that possess an S-shaped steady-state response curve (i.e. with multiple output values for certain input values) and means that the system may settle in two different stable states depending on the initial conditions (Fig 1A). These observable stable steady states correspond to the lower and top branch of the S-curve, while the middle part of the S-curve is unstable and cannot be measured experimentally. The full S-shaped response curve can therefore not be obtained experimentally, and bistability typically manifests itself through sharp jumps and hysteresis in the measurements. Hysteresis appears when the threshold for switching from low to high response levels is different from the threshold for switching from high to low response levels. In the cell cycle, these all-or-none responses ensure robust transitions between different cell cycle phases, while hysteresis prevents the cell cycle from returning to previous phases without having been through the whole cell cycle. Control mechanisms at checkpoints can prevent such transitions, either by keeping the input at sub-threshold levels or by shifting the right threshold to higher input levels [4–6].

One of the first cell cycle transitions for which the role of an underlying bistable switch was established, is mitotic entry in early embryonic cells of *Xenopus* frogs. This cell cycle transition is characterized by the switch-like phosphorylation of numerous proteins, referred to as mitotic substrates. Throughout interphase, cyclin B (CycB) molecules are gradually produced and bind to cyclin-dependent kinase 1 (Cdk1). At the onset of mitosis, the phosphorylation state of these CycB-Cdk1 complexes abruptly changes, resulting in their switch-like activation and subsequent phosphorylation of the mitotic substrates [7]. The sudden changes in phosphorylation state of CycB-Cdk1 are generated by positive and double negative feedback loops with the phosphatase Cdc25 and kinase Wee1, respectively [8]. Theoretical work proposed that bistability, resulting from these feedback loops, plays an important role in cell cycle progression [9, 10]. Afterwards, this was experimentally validated [11–13]. Not only CycB-Cdk1, but also counteracting phosphatases such as PP2A help in regulating the phosphorylation state of mitotic substrates, and thus entry into and exit from mitosis. Recent findings showed how PP2A too is regulated in a bistable manner [14–16]. Furthermore, the phosphatases PP1 and

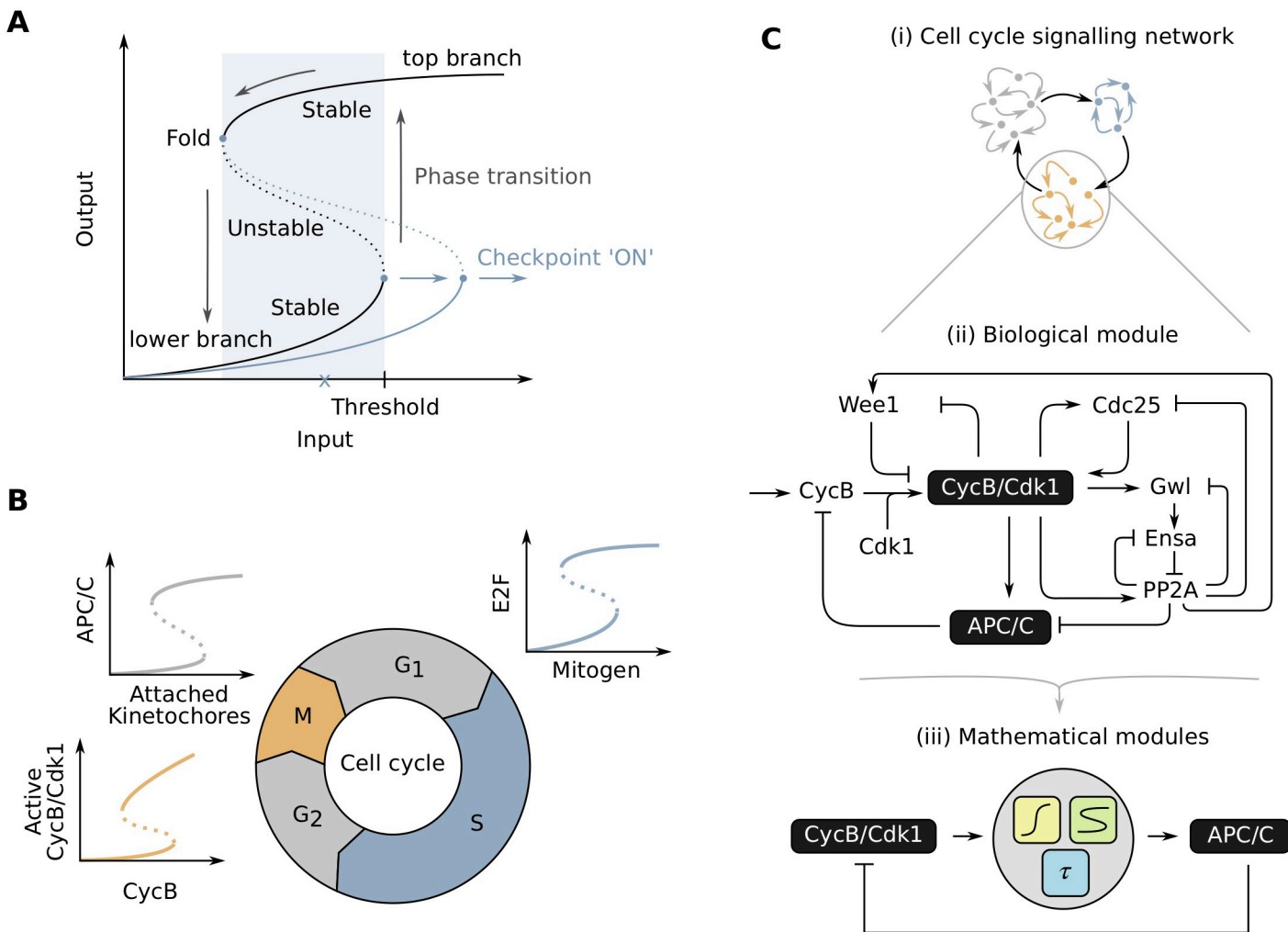

**Fig 1. Modularity in the cell cycle. (A)** An S-shaped response is characterized by the coexistence of two stable and one unstable steady states for a certain range of input levels (blue shaded area). Cell cycle transitions can be described as sudden jumps from the lower to upper branch of the curve (or vice versa). Checkpoints act by fixing the input at sub-threshold levels (x) or shifting the threshold value to a different input level (blue curve). **(B)** Different phase transitions in the cell cycle have been shown to be centered around bistable switches for which S-shaped response curves have been experimentally measured. **(C)** Entry into mitosis (and other cell cycle transitions) can be isolated as a biological module from the bigger network of other cell signaling events in the cell (i). Whereas the dynamical features of these modules have been studied extensively by considering detailed reaction schemes (ii), we propose a phenomenological approach based on three easily combinable 'functional modules': time delay, ultrasensitive and S-shaped responses (iii).

Fcp1 have been implicated in regulating the exit from mitosis, but how exactly these phosphatases interact with the different cell cycle regulators remains a topic of active research [17, 18].

The observation that, just like M phase entry, other phase transitions of the cell cycle are irreversible under normal physiological conditions, suggested that these too are built on bistable switches. For example, the transition from G1 to S phase is governed by a complex interplay between extracellular signals, CycD-Cdk4/6, the transcription factor E2F, retinoblastoma protein (Rb) and CycE-Cdk2. These interactions lead to bistability in the activity of E2F [19]. Appropriate conditions, such as a sufficient concentration of extracellular growth factors and nutrients, will push the cell across the threshold of the switch to a 'high E2F' state. At this point the cell irreversibly commits to the cell cycle, i.e. it will finish the started round of cell division even if nutrient levels drop [5, 20]. Another cell cycle transition for which bistability has been proposed to fulfill an important role is the metaphase-to-anaphase transition

and the accompanying spindle assembly checkpoint (SAC). During this transition, microtubules in the mitotic spindle need to correctly attach to the sister chromatids. Once these are correctly attached, the cohesin rings that are keeping the sister chromatids together can be cleaved, upon which the chromatids are separated by the mitotic spindle. Although some experimental studies question the all-or-none nature of the SAC [21, 22], indirect experimental and theoretical findings support the idea that this transition is also centered around a bistable switch [23–26]. Given the recurring occurrence of all-or-none transitions throughout the cell cycle, the latter has been envisioned as a chain of interlinked bistable switches (Fig 1B) [26, 27].

Although a chain of bistable switches can account for control mechanisms at cell cycle transitions and checkpoints, an additional mechanism is still needed to drive the cycle forward and reset it back to its initial state at the end—thus putting the dominoes back up after toppling them. This is provided by the periodic production and degradation of cyclins [28]. The CycB-Cdk1 complexes that accumulate during interphase are activated at mitotic entry. In turn, they will activate the Anaphase-Promoting Complex/Cyclosome (APC/C), a ubiquitin ligase. This protein complex then induces the degradation of the cyclins [29]. APC/C regulation is believed to be a complex multi-step mechanism in which time delays play an important role, hence introducing a delayed negative feedback loop in the reaction network [30, 31]. The latter is generally known to allow for robust and sustained oscillatory behavior of dynamical systems [32]. Even when the cellular checkpoints are absent, this negative feedback loop can drive autonomous biochemical oscillations in a 'clock-like' manner, as for example seen in *Xenopus* and sea urchin eggs which rapidly alternate between phases of DNA replication (S phase) and segregation of the chromosomes (M phase) [3, 33].

The cell cycle is a complex process with many interacting parts. Dividing it into a sequence of discrete modules such as bistable switches might seem artificial. However, since its introduction in biology by Hartwell *et al.*, [34], such a modular approach describing biological processes has been justified by advances in synthetic biology, genomics, cell signalling and single-cell techniques [35–38]. Furthermore, studying the cell cycle based on discrete modules is warranted by the temporal segregation of the different cell cycle phases, and the presence of bistability itself [39, 40].

Even though separating the full system into different modules greatly reduces the complexity, understanding the dynamical behavior of those modules often requires mathematical models instead of mere intuition [41]. Even for a single module, a biochemically detailed study results in a large number of variables and parameters, many of which are difficult to determine experimentally. As this might hamper interpretation of the results, it is often desirable to reduce the complexity of the mathematical model. One way to accomplish this is by focusing on a core subnetwork and omitting all other reactions, thus (hopefully) capturing the key qualitative behavior of the system [42]. Although this approach certainly has its merits, some dynamical properties such as bistability may be lost when reducing the model too much [43, 44]. Of note, this does not mean that models of small reaction networks imply less interesting dynamical features: for example, even a system containing a single molecule can behave in a bistable manner [45].

Another strategy to reduce the complexity of a model, while retaining much of the system's dynamical behavior, is by making certain simplifying assumptions about the underlying reactions, rather than omitting them. For example, using timescale separation methods, one can identify variables that evolve on a much faster timescale than others. The equations for these variables can be replaced by expressions for the steady-state response curves, which can be introduced into the equations for the slow variables. This process is often responsible for the introduction of nonlinearities in a reaction system. Examples include Michaelis-Menten

enzyme kinetics, zero-order ultrasensitivity, multisite phosphorylations, cooperative binding events and stoichiometric inhibition [46], whose net result is often summarized by their steady-state response curve. These steady-state response curves can be described by reaction-specific formulas, such as the Michaelis-Menten equation for enzyme-kinetics [47] or the Goldbeter-Koshland function for zero-order ultrasensitivity [48]. Alternatively, steep responses —of different molecular origins— can often be adequately approximated by a Hill function [49–51]. The Hill function is therefore used to introduce steep responses, without regarding the underlying reactions that generated them. Another example of dynamical behavior that can be explicitly incorporated are time delays, which arise in biochemical reactions due to the non-zero time to complete physical and/or biochemical processes, such as molecular transport or intermediate reactions [52, 53]. Such delays can be accounted for via delay differential equations. In addition to being simpler than mechanistic models, a phenomenological model based on explicit mathematical expressions is often the sole option to describe experimentally observed responses whenever the underlying molecular mechanisms are unknown.

S-shaped responses typically emerge from regulatory mechanisms like positive or double negative feedback loops [54]. Many cell cycle models include these feedback loops to generate the S-shaped response [10, 55–57]. Steep responses and time delays have been explicitly included in cell cycle models using a simple mathematical form. Given the recurring occurrence of bistable switches in the cell cycle, it is remarkable that such a direct mathematical formulation of S-shaped response curves has never been, as far as we know, explicitly incorporated into cell cycle models. Here we want to close this gap (Fig 1C) by providing an easily tunable, phenomenological expression for such an S-shaped response. This expression can easily be converted into a functional module for ultrasensitivity by tuning a few parameters, and can be combined with a time delay. As such, we provide a toolbox of three functional modules (ultrasensitive response, S-shaped response, time delay) that can be combined in different configurations to model a variety of biological processes. We will illustrate this approach with models that include different bistable switches in the cell cycle.

## Results

### Ultrasensitive responses, S-shaped responses and delay as functional modules

Before considering actual models of the cell cycle, we start with a general overview of the functional modules that will be used in the rest of the paper: ultrasensitivity, S-shaped response and delay. Nonlinear response curves are often described by a Hill function of the form Output $= \frac{\text{Input}^n}{K^n + \text{Input}^n}$. For $n > 1$, this gives a sigmoidal response curve, with higher values of $n$ corresponding to steeper responses (Fig 2A). Such steep responses are often said to be 'ultrasensitive' [51]. Our ultrasensitive module thus consists of a Hill function.

In order to produce a mathematical expression for our second module, the S-shaped response, we would like to 'bend' the ultrasensitive curve just described such that it becomes S-shaped (Fig 2B). Mathematically, this can be achieved by starting from a Hill function, and multiplying the threshold value $K$ with a scaling function $\xi(\text{Out})$:

$$\text{Out} = \frac{\text{In}^n}{K^n + \text{In}^n} \qquad \Rightarrow \qquad \text{Out} = \frac{\text{In}^n}{[\xi(\text{Out}) \cdot K]^n + \text{In}^n}. \qquad (1)$$

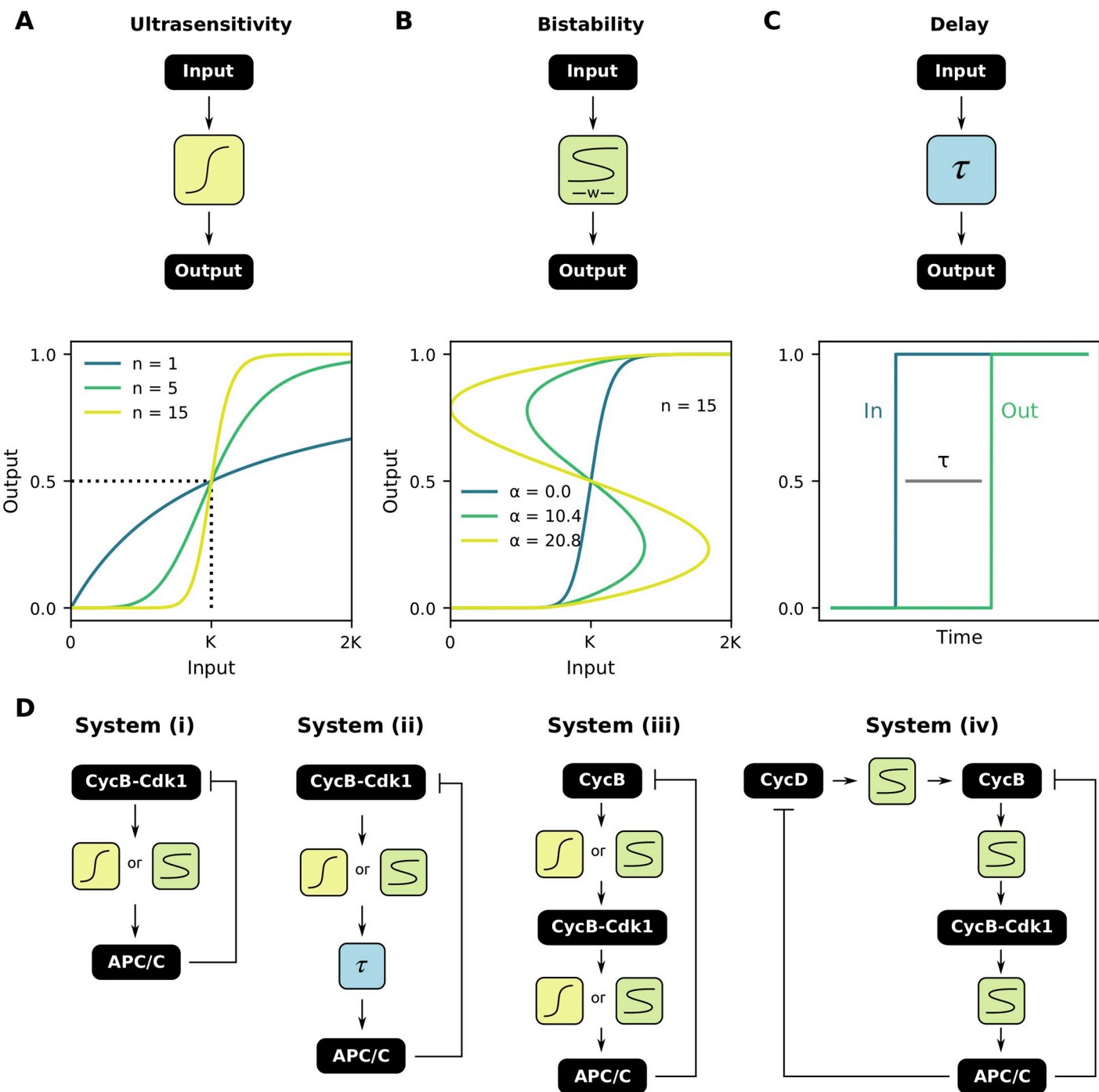

**Fig 2. Ultrasensitive responses, S-shaped responses and delay as functional modules.** (**A**) An ultrasensitive input-output response is characterized by the threshold value $K$ and the Hill exponent $n$. (**B**) An ultrasensitive function can be converted into an S-shaped response of different widths: for $\alpha = 0$, an ultrasensitive response is obtained, while increasing $\alpha$ increases the width of the S-shaped region. These plots were generated by calculating the inverse of Eq 1. (**C**) A certain lag time $\tau$ between input and output is encountered in several biochemical reactions. (**D**) Several combinations of the three functional modules, always in the presence of negative feedback, to describe oscillations in the cell cycle.

Note that the latter expression does not define a function—which is impossible, as per definition an S-shaped response has multiple outputs for one input. Instead, it should be interpreted as the steady-state response of an ordinary differential equation (ODE):

$$\frac{d\text{Out}}{dt} = \frac{\text{In}^n}{[\xi(\text{Out}) \cdot K]^n + \text{In}^n} - \text{Out}. \tag{2}$$

Although different options for $\xi$ are possible (see S1 Fig), a straightforward choice in analogy with, for example, the FitzHugh-Nagumo model would be a cubic function [58]:

$$\xi(\text{Out}) = 1 + \alpha \cdot \text{Out}(\text{Out} - 1)(\text{Out} - r).$$

Changing the parameter $\alpha$ allows for a smooth transition between an ultrasensitive and S-shaped response: for $\alpha = 0$ we retrieve the original ultrasensitive response, while increasing its value results in wider S-shaped regions (Fig 2B). The parameter $r$ can be used to alter the symmetry of the response curve. In the remainder of the text, we keep $r = 0.5$ (resulting in a symmetric response as in Fig 2B), and we will not interpret the parameter $r$ further biologically. The effect of the different parameters on the shape of the scaling function is discussed in more detail in S1 Text and S2 Fig.

For the last module, i.e. delay, a lag time $\tau$ between input and output of the system (Fig 2C) can be introduced. Replacing In($t$) by In($t - \tau$) then converts Eq 2 into a delay differential equation. In what follows, we will combine these three functional modules in several configurations, always in combination with a negative feedback (Fig 2D), to model different aspects of the cell cycle. We will explain how changing the mathematical properties and function parameters influence the overall dynamical behavior of the model.

## S-shaped, but not ultrasensitive, responses cause a two-dimensional system of the cell cycle to oscillate

The most straightforward strategy for combining the different functional modules into actual cell cycle models, would be to start with a simple—but biologically relevant—reaction network and gradually add additional modules to increase the scope of the model. The simplest cell cycle models arguably describe the 'clock-like' cell cycle oscillations in *Xenopus laevis* eggs, compared with the more complex 'domino-like' mechanism in somatic cells. Indeed, the early embryonic cell cycle in *X. laevis* simply cycles between S phase and M phase and lacks checkpoints and gap phases [33]. Furthermore, cell cycles 2 till 12 after fertilization of the egg are characterized by an increased activity of the phosphatase Cdc25 relative to the kinase Wee1, resulting in quick activation of CycB-Cdk1 and subsequent APC/C activation [59]. Taken together, the reaction network at the core of the early embryonic cell cycle can be simplified by a negative feedback loop, where active CycB-Cdk1 phosphorylates and activates APC/C, which then causes degradation of CycB molecules. A two-variable phenomenological model of this system is given by (Fig 3A):

$$\begin{cases} \dfrac{d[\text{Cdk1}]}{dt} &= b_{\text{syn}} - b_{\text{deg}}[\text{Cdk1}] \cdot [\text{APC}]^* \\[2mm] \dfrac{d[\text{APC}]^*}{dt} &= \dfrac{1}{\epsilon_{\text{apc}}}\left( \dfrac{[\text{Cdk1}]^n}{K_{\text{cdk,apc}}^n + [\text{Cdk1}]^n} - [\text{APC}]^* \right) \end{cases}. \tag{system(i-a)}$$

Here, the two variables [Cdk1] and [APC]$^*$ represent the concentrations of activated CycB-Cdk1 complexes and the ratio of activated APC/C to total APC/C, respectively. The first

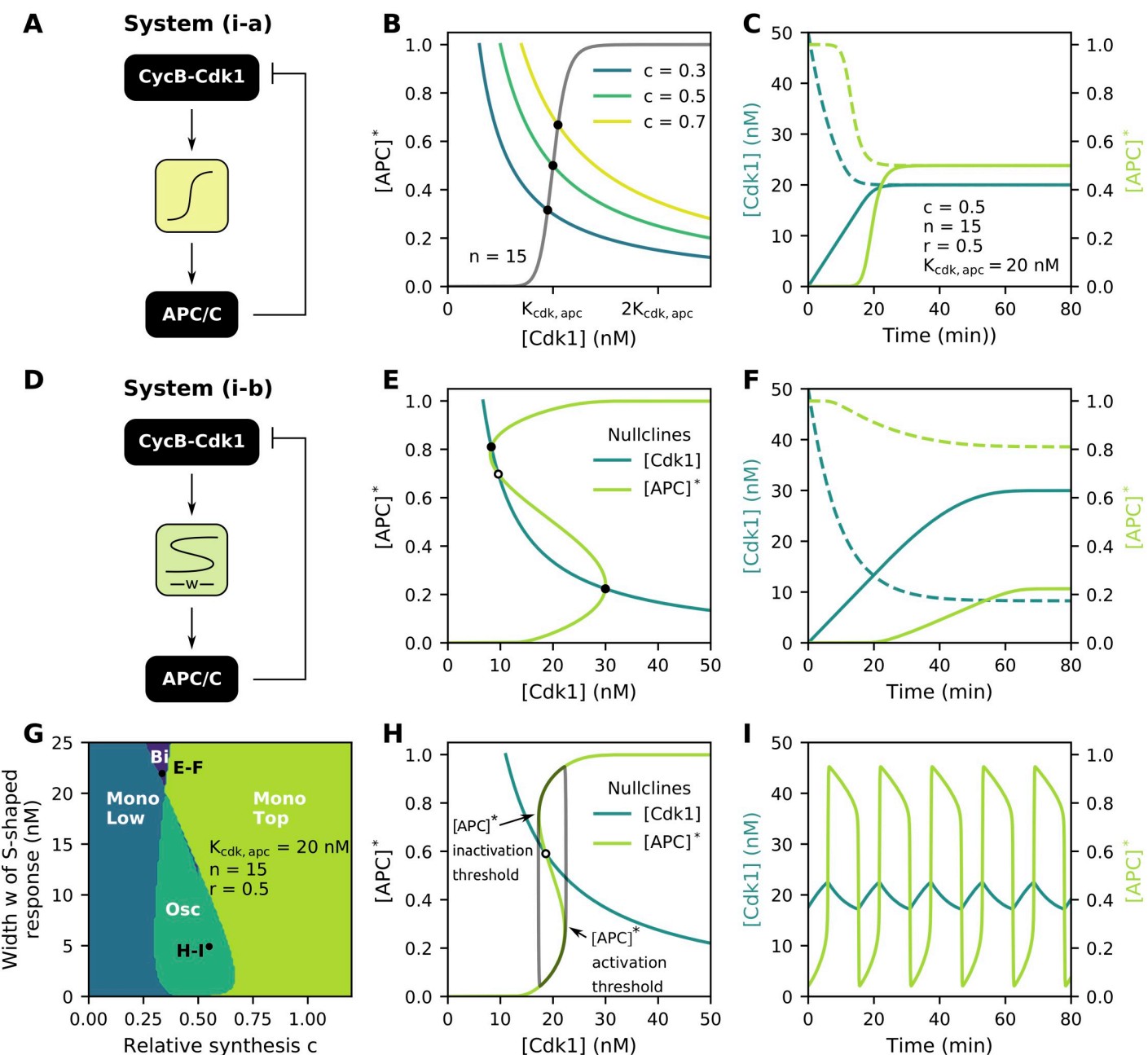

**Fig 3. S-shaped, but not ultrasensitive, responses cause a two-dimensional system of the cell cycle to oscillate.** Parameter values as indicated in the figure. **(A)** Block diagram of the ultrasensitive, negative feedback system (i-a). **(B)** Effect of relative synthesis *c* on the location of the [Cdk1] nullcline in the phase plane. **(C)** Time traces for two different initial conditions of system (i-a), denoted by either continuous or dotted line. Only one steady state exists. **(D)** Block diagram of the bistable, negative feedback system (i-b). **(E)** For wide S-shaped [APC]* nullclines, three intersections in the phase plane can exist, resulting in bistability of the overall ODE system. Closed black dots depict stable steady states, while open dots denote unstable steady states. **(F)** Time traces for system (i-b). Depending on the initial conditions, the system will settle in one of two stable steady states. **(G)** Number and position of the steady states for system (i-b). Osc = oscillations, Bi = bistable, Mono Low = one stable steady state on lower branch of [APC]* nullcline, Mono Top = one stable steady state on top branch of [APC]* nullcline. **(H-I)** If the [Cdk1] nullcline intersects the [APC]* nullcline in between its folds, one unstable steady state exists, resulting in oscillations.

equation describes the rate of change of activated CycB-Cdk1 complexes and consists of two terms: the constant synthesis of cyclins and their APC/C-dependent degradation. In this model all synthesized cyclin immediately binds to Cdk1 and activates it. This is why cyclin synthesis is directly included in the equation for [Cdk1]. The second equation in system (i-a)

states that the rate of change of $[APC]^*$ is proportional to the difference between the experimentally determined $[APC]^*$ ultrasensitive steady-state response and its current level (similar as in Eq 2) [30, 59], which it approaches with a rate constant $1/\epsilon_{apc}$. Note thus how positive and negative terms in this ODE are not to be interpreted as actual production and degradation terms, respectively.

Because system (i-a) has two variables, its behavior can be conveniently studied in the phase plane. To determine the steady-state behavior of system (i-a), one needs to find the number and location of intersections of the $[Cdk1]$ and $[APC]^*$ nullclines. These are the curves in the $[Cdk1]$-$[APC]^*$ phase plane for which $d[Cdk1]/dt = 0$ and $d[APC]^*/dt = 0$ respectively. Here, only one intersection can exist, whose location is determined by the relative position of both nullclines (Fig 3B and 3C). Moreover, this steady state is stable, as seen from the linearized system (see S1 Text for details). While parameters $n$ and $K_{cdk,apc}$ affect the ultrasensitive $[APC]^*$ nullcline, the location of the $[Cdk1]$ nullcline is determined by the ratio of $b_{syn}$ and $b_{deg}$. However, after non-dimensionalizing the system to facilitate mathematical analysis (see Materials and methods for details), this ratio can be re-expressed as the product of $K_{cdk,apc}$ with a newly introduced dimensionless parameter $c$:

$$c = \frac{b_{syn}}{K_{cdk,apc} \cdot b_{deg}}.$$

In what follows, we will refer to the cyclin production terms in the non-dimensionalized ODEs as the 'relative synthesis rate', which for system (i-a) (and system (i-b) and system (ii) below) is given by parameter $c$. Changing $c$ changes the location of the $[Cdk1]$ nullcline in the phase plane: whereas low values shift it to the left, high $c$ values shift the $[Cdk1]$ nullcline to the right (Fig 3B). It is mainly this parameter that will be used in the remainder of the text when assessing the effect of the $[Cdk1]$ nullcline on the dynamical behavior of the overall system.

As the steady state is stable, this system can not serve as a basic model to describe cell cycle oscillations. To make the system oscillate, either a time delay or an S-shaped response can be introduced. The effect of a time delay in combination with ultrasensitivity has previously been studied in detail [60] (see also S3 Fig). Here we will focus on the effect of converting the ultrasensitive module from system (i-a) into an S-shaped module, which is in line with recent experimental findings that showed the existence of hysteresis in APC/C response curves [15]. As discussed above, the threshold $K_{cdk,apc}$ can be multiplied with a cubic scaling function $\xi([APC]^*) = 1 + \alpha_{apc}[APC]^*([APC]^* - 1)([APC]^* - r)$ (Fig 3D):

$$\begin{cases} \dfrac{d[Cdk1]}{dt} = b_{syn} - b_{deg}[Cdk1] \cdot [APC]^* \\[2mm] \dfrac{d[APC]^*}{dt} = \dfrac{1}{\epsilon_{apc}} \left( \dfrac{[Cdk1]^n}{(\xi([APC]^*) \cdot K_{cdk,apc})^n + [Cdk1]^n} - [APC]^* \right) \end{cases} \quad \text{(system(i-b))}$$

Note that by multiplying the threshold $K_{cdk,apc}$ with $\xi$ in system (i-b), the $[APC]^*$ nullcline becomes S-shaped when $\alpha_{apc} > 0$ (as introduced in Eq 2 and Fig 2B). Systems exhibiting such an S-shaped response are good candidates for showing bistability, in which case the system can evolve over time to two steady states. Whether this behavior is indeed observed, depends on the number of intersections between the $[Cdk1]$ and $[APC]^*$ nullclines in the two-dimensional phase plane. As the $[APC]^*$ nullcline is now S-shaped instead of sigmoidal, either one or three intersections can exist with the $[Cdk1]$ nullcline (Fig 3E, 3G and 3H). The system is bistable when three intersections exist, with one steady state being unstable and the other two stable. It is the initial condition of the system that determines in which of

the two stable ones the system will settle (Fig 3F). Whenever the system has only one steady state (i.e. one intersection of the nullclines), the latter can either be stable (on the upper or lower branch of the S-shaped nullcline) or unstable. Only in the latter case, which happens if $\epsilon_{apc}$ is small enough and both nullclines intersect in between the fold points of the [APC]$^*$ nullcline (Fig 3H), sustained oscillations around this steady state appear. These oscillations can be thought of as 'orbiting around the nullclines', i.e. they show increasing [Cdk1] levels on the lower branch of the response curve until the [APC]$^*$ activation threshold is reached. At this point, [APC]$^*$ levels jump to a higher value, which triggers the decrease of [Cdk1]. The system then proceeds along the upper branch until the inactivation threshold is reached and the cycle is complete. These oscillations, characterized by slow progress along the branches of the nullclines and quick jumps between them, are called relaxation oscillations. They are similar to oscillations observed in, for example, the FitzHugh-Nagumo equations [58].

From the observation that the system oscillates if the nullclines have a single intersection in the middle, we can derive some qualitative conditions for oscillations to exist: the [APC]$^*$ null-cline should not be too wide and the [Cdk1] nullcline should not be located too far to the left nor to the right (relative to the other nullcline). Indeed, from Fig 3G, we see that oscillations are favored for $c$ values around 0.5 and narrow S-shaped regions. As the S-shaped region becomes wider, the period of oscillations and [Cdk1] amplitude increases (S4 Fig), but the range of relative synthesis $c$ for which oscillations can be sustained decreases. For [APC]$^*$, the effect of the width of the S-shaped region on the amplitude is more moderate, and the amplitude is near-maximal for most oscillations. From the time profiles, we further see that the overall shape of the oscillations has a typical sawtooth-like waveform (Fig 3I). It is interesting to note that, in this system, the effect of the S-shaped response is mixed: on the one hand, it is required for the system to oscillate, but on the other hand a wider S-shaped region makes oscillations less likely, as in that case the system settles in a steady state on the upper or lower branch.

## The S-shaped module reproduces the behavior of mass-action models

In the previous section, we showed how the S-shaped module can be used to conveniently model oscillatory systems and showed that the obtained oscillations resemble those of the early embryonic cell cycle in a qualitative way. The way we introduced the S-shaped response, through the modified Hill function, is artificial, and not based on biochemical interactions. To show that our approach can represent the behavior of an actual biochemical system, we compare our module-based system to an existing cell cycle model based on mass-action kinetics. For this purpose, we first extended a previously described mass-action model of the PP2A-ENSA-GWL network [61], by incorporating synthesis and APC/C-mediated degradation of CycB (Fig 4A). This system is known to generate S-shaped APC/C response curves via a double negative feedback loop. More specifically, GWL (which is phosphorylated by CycB-Cdk1) indirectly inhibits PP2A-B55 by phosphorylating ENSA, which is both substrate and inhibitor of PP2A [62]. PP2A itself, when active, dephosphorylates and inactivates GWL, thus closing the double negative feedback loop. This leads to an S-shaped response of PP2A activity as function of CycB-Cdk1 activity. As APC/C is a substrate of PP2A, the S-shaped steady-state response of PP2A is translated into a similar response for APC/C as function of CycB-Cdk1.

Once the S-shaped steady-state response curve of the mass-action model (for a fixed parameter set, Table 1 in Materials and methods) was calculated, we approximated this response

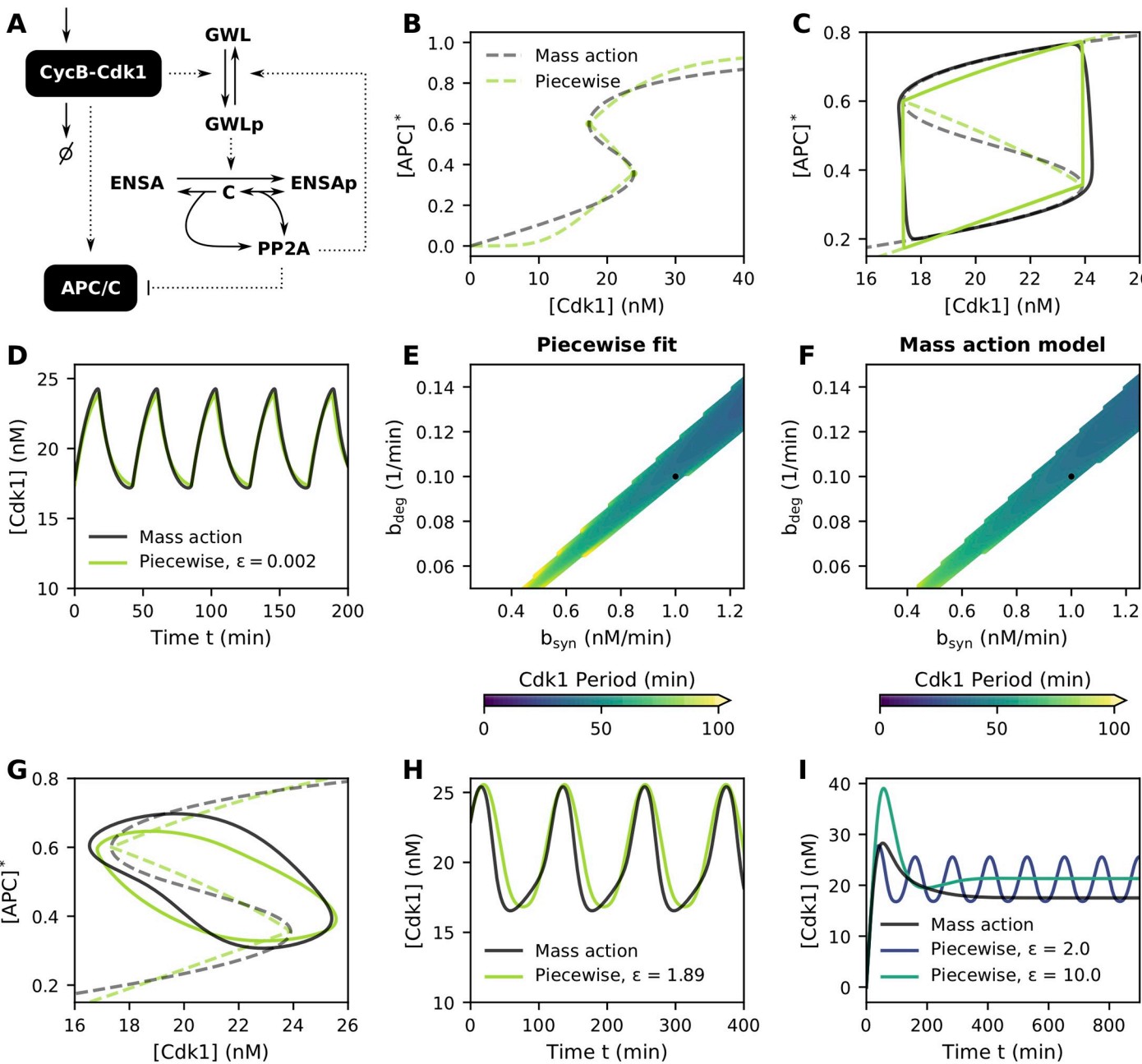

**Fig 4. The S-shaped module reproduces the behavior of mass-action models.** Parameter values as given in Table 1 or as indicated in the figure/caption. **(A)** Mass-action model of the PP2A-ENSA-GWL network in which double negative feedback can lead to an S-shaped response of APC/C. **(B)** Steady-state response curve for mass-action model and fitted S-shaped module. **(C,D)** Time traces of the system, in the phase plane (C) and time domain (D). Note how the time traces closely follow the steady-state response curve. **(E,F)** Oscillatory region as a function of CycB synthesis and degradation rates for the piecewise fit and mass-action model. Black dot denotes values used for the time traces in panels C,D,G,H,I. **(G,H)** Time traces of the system, in the phase plane (G) and time domain (H). All mass-action rate constants (except from synthesis and degradation) from Table 1 were divided by a constant factor (i.e. 16.5), leading to less separated timescales and reduced correspondence between the time traces of both models. **(I)** By further decreasing the mass-action parameters from panels G,H (i.e. factor 100), the correspondence between the mass-action and phenomenological system becomes even worse, irrespective of the choice of $\epsilon$.

using a piecewise linear scaling function $\xi([APC]^*)$ (Fig 4B and S1 Fig, see Materials and methods for details). Using the same cyclin synthesis and degradation constants for both models, the oscillatory period of the phenomenological model can be adjusted to match that of the mass-action model by simply changing the time constant $\epsilon$ (Fig 4D). From Fig 4C, it is seen how the time traces stay close to their steady-state response curve. Accordingly, the phenomenological model will approach the mass-action-model as good as one can fit the steady-state response. Because the synthesis and degradation rates of CycB do not affect the steady-state curve, their values can be changed while maintaining a good correspondence between both models (Fig 4E and 4F).

As alluded to above, a necessary condition for letting the phenomenological model correspond to the mass-action model by fitting the steady-state responses, is the close approximation of the time traces to the steady-state curves. In the example just given, this was achieved by choosing the synthesis and degradation rates smaller than the other reaction rates in the mass-action model [13]. However, the assumption that the timescale of synthesis and degradation is well separated from that of the other reactions in the system is not necessarily a valid one [13]. When this condition is not satisfied, the phenomenological approach of fitting the steady-state response curves can lead to inaccurate results. Indeed, when scaling all reaction rates (except from synthesis and degradation) in the mass-action model such that the timescales are less clearly separated, the trajectory in the phase plane does no longer follow the outer branches of the S-shaped curve (Fig 4G). As such, a good fit of the latter does not guarantee a good correspondence of the time traces (Fig 4H). As an additional example, consider Fig 4I, where the mass-action system settles in a steady state, whereas the phenomenological approximation either oscillates or settles in a different steady state.

It should be emphasized that similar validity conditions apply when using the ultrasensitive module, as here too, the system only approximates the steady-state response curve when the timescales are well separated. In fact, such timescale separation is the basis for other model reduction and approximation methods (e.g. the well-known quasi steady-state approximation), which too are valid only under the right circumstances [63]. Our method is different from others in that we substitute the fast dynamics by a manually tunable response curve which is not directly derived from the original equations.

## Delay increases the period, amplitude and robustness of oscillations

So far, we have described the regulation of APC/C by CycB-Cdk1 as a module in which changes in CycB-Cdk1 immediately affect APC/C activity. In reality, however, this regulation is believed to be a complex multi-step mechanism in which time delays play an important role [30]. The effects of these delays can be modeled in a phenomenological way by using delay differential equations. Here, we will focus on a basic example with two types of delays, but it should be noted that the way a delay is implemented in a model can drastically affect the outcome [60]. First, a delay $\tau_1$ between the activation of CycB-Cdk1 and the subsequent activation of APC/C can exist, i.e. a lag time between [Cdk1] levels reaching the right fold of the $[APC]^*$ nullcline and $[APC]^*$ levels actually jumping to the upper branch (Fig 5C, 5D, 5E and 5F). Secondly, a delay $\tau_2$ might occur between the inactivation of CycB-Cdk1 and APC/C. The delays $\tau_1$ and $\tau_2$ can either be the same ($\tau = \tau_1 = \tau_2$) or they can take different values (if for example, in the biological system the activation of APC/C were to take much longer than its inactivation, delay $\tau_1$ would have a larger value than $\tau_2$ [60]). The case where $\tau_1 \neq \tau_2$ can be implemented into the mathematical equations by expressing the delay $\tau$ as a function of the $[APC]^*$ levels, so that $\tau = \tau_1$ when APC/C activity is low (i.e. $[APC]^* < 0.5$) and $\tau = \tau_2$ when APC/C

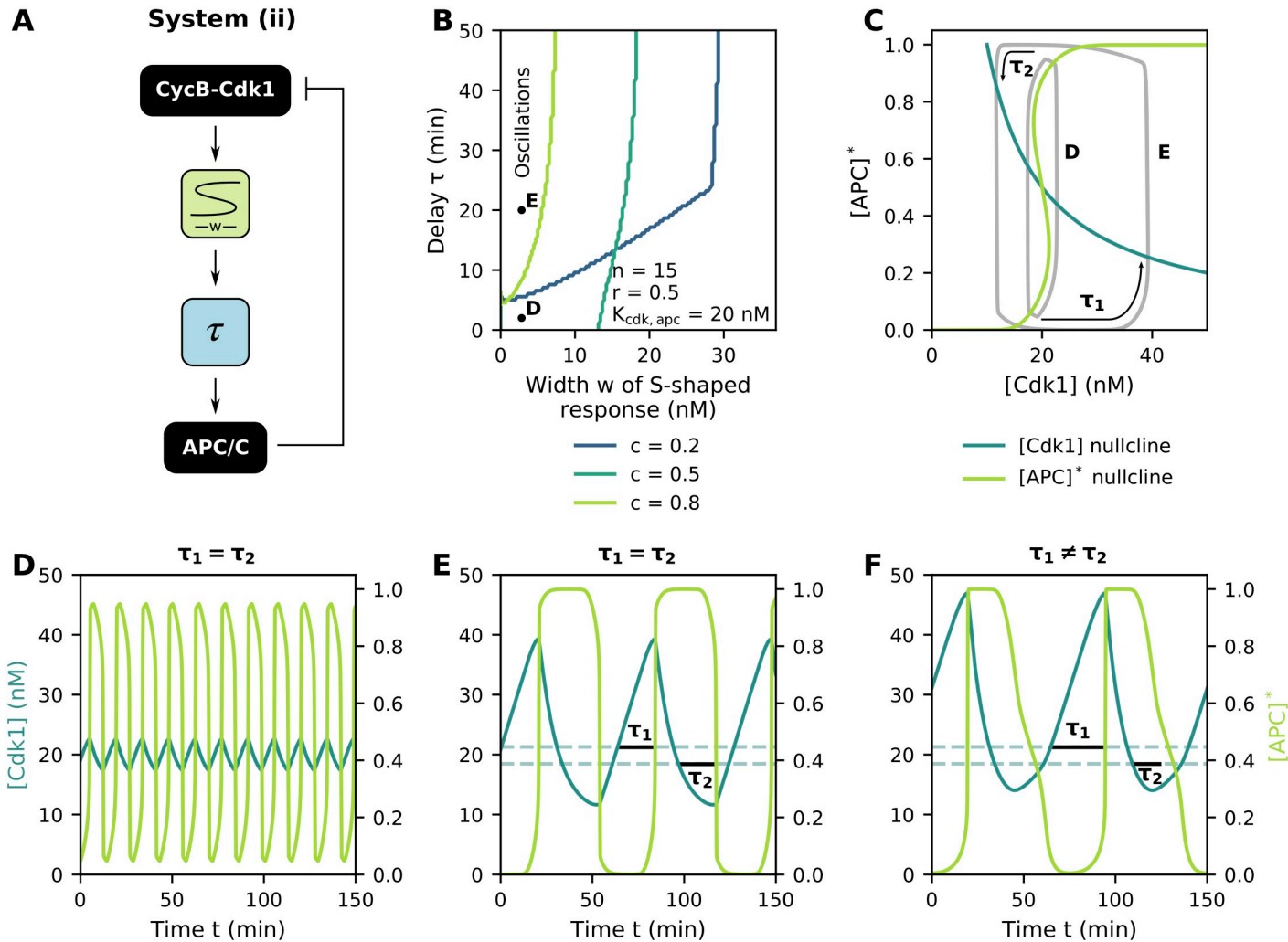

**Fig 5. Delay increases the period, amplitude and robustness of oscillations.** Parameter values as indicated in the figure/caption. **(A)** Block diagram of the bistable, delayed negative feedback network. **(B)** Oscillatory regions, located above and to the left of the plotted lines, as a function of the time delay and width of the S-shaped response for different values of the relative synthesis rate $c$. **(C)** Phase plane representation of the oscillations for different values of $\tau$. Note how one can only conclude that $\tau_1 = \tau_2$ from time traces, not directly from the phase plane. **(D,E)** Time traces for parameter values indicated in panel B and $c = 0.5$. $\tau_1$ denotes the delay between activation of [Cdk1] and activation of [APC]$^*$, while $\tau_2$ represents the delay between their inactivations. Dashed lines depict the [Cdk1] levels at the folds of the bistable [APC]$^*$ nullcline. **(F)** State-dependent time delays can account for differences in the activation and inactivation delays ($c = 0.5$, $p = 5$).

activity is high (i.e. [APC]$^* > 0.5$). Taken together, we arrive at the following system:

$$\begin{cases} \dfrac{d[\text{Cdk1}]}{dt} & = b_{\text{syn}} - b_{\text{deg}}[\text{Cdk1}] \cdot [\text{APC}]^* \\[2mm] \dfrac{d[\text{APC}]^*}{dt} & = \dfrac{1}{\epsilon_{\text{apc}}}\left( \dfrac{[\text{Cdk1}]^n(t-\tau)}{(\xi([\text{APC}]^*) \cdot K_{\text{cdk,apc}})^n + [\text{Cdk1}]^n(t-\tau)} - [\text{APC}]^* \right) \end{cases} \qquad (\text{system(ii)})$$

$$\text{with } \tau = \tau_1 = \tau_2$$

$$\text{or } \tau = \tau([\text{APC}]^*) = \tau_1 + (\tau_2 - \tau_1)\frac{[\text{APC}]^{*p}}{0.5^p + [\text{APC}]^{*p}}.$$

Here, the Hill function in $\tau([APC]^*)$ approaches a step function (for high values of the Hill exponent $p$) with $\tau \approx \tau_1$ if $[APC]^*$ is smaller than 0.5 (as the Hill function approaches zero) and $\tau \approx \tau_2$ when $[APC]^*$ activity is larger than 0.5 (as the Hill function approaches one), as is desired (recall that $\tau_1$ represents the 'activation delay' and $\tau_2$ the 'inactivation delay').

For $\xi([APC]^*) = 1$ (i.e. $\alpha_{apc} = 0$) we retrieve the combination of a delay and ultrasensitive module (S3 Fig). As before, increasing $\alpha_{apc}$ converts the ultrasensitive response into an S-shaped one. Without a time delay, oscillations can be observed for intermediate values of the relative synthesis $c$ ($c = 0.5$) whenever the S-shaped region is not too wide (as otherwise the overall system becomes bistable). Low or high $c$ values result in one stable steady state and no oscillations (compare Figs 5B and 3G). However, adding a sufficiently large time delay $\tau$ can cause the system to start oscillating, even for widths of the S-shaped region and values of $c$ where the system is mono- or bistable if $\tau = 0$. The oscillator thus becomes more robust against changes in the width of the S-shaped response curve. The wider the S-shaped region becomes, the larger the delay required to induce oscillations. A maximal width exists beyond which no oscillations can be sustained, independent of the time delay, corresponding to the right vertical boundary of the oscillatory region in Fig 5B (especially clear for $c = 0.2$).

Both the period and [Cdk1] amplitude grow as the time delay increases (Fig 5C, 5D, 5E and S5 Fig). Activation of $[APC]^*$, on the other hand, is an all-or-none process for the majority of parameter values, with longer delays prolonging the time during which the $[APC]^*$ activity is at its maximal/minimal level, resulting in plateau phases in the time profiles (Fig 5C, 5D, 5E and S5 Fig).

## Large amplitude oscillations are facilitated by two bistable switches

In the previous section we considered CycB-Cdk1 activity to be equivalent to CycB levels. This is justified for cycles 2–12 of the embryonic cell cycle of *X. laevis*, where all CycB quickly binds to Cdk1 and the complexes are rapidly activated by the phosphatase Cdc25 whose activity is dominant over the inhibitory kinase Wee1 [59]. However, in the first embryonic cycle and extracts of *Xenopus* eggs, a bistable response of CycB-Cdk1 activity with respect to total CycB concentrations is typically observed [11, 12, 59]. Hence, a logical next step in adjusting our cell cycle model is to add a separate S-shaped module (or as it will appear, a slightly modified version of it) for this regulation (Fig 6A):

$$\begin{cases} \dfrac{d[CycB]}{dt} &= b_{syn} - b_{deg}[CycB] \cdot [APC]^* \\[2em] \dfrac{d[Cdk1]}{dt} &= \dfrac{1}{\epsilon_{cdk}}\left( \dfrac{[CycB]^n}{\left[\xi\left({}^{[Cdk1]}\big/_{[CycB]}\right) \cdot K_{cyc,cdk}\right]^n + [CycB]^n}[CycB] - [Cdk1] \right) \cdot \\[2em] \dfrac{d[APC]^*}{dt} &= \dfrac{1}{\epsilon_{apc}}\left( \dfrac{[Cdk1]^n}{(\xi([APC]^*) \cdot K_{cdk,apc})^n + [Cdk1]^n} - [APC]^* \right) \end{cases} \quad \text{(system (iii))}$$

Here, the variable [CycB] represents the total concentration of CycB molecules, [Cdk1] represents activated CycB-Cdk1 complexes and $[APC]^*$ the ratio of active APC/C molecules to the total amount of APC/C. Note how for the rate of change of [Cdk1], the S-shaped module is slightly modified compared with the one used in system (i-b): the Hill-like expression for the S-shaped response is multiplied by [CycB]. The reason for this is explained in detail in Materials and methods, and for now, it suffices to mention that this multiplication does not

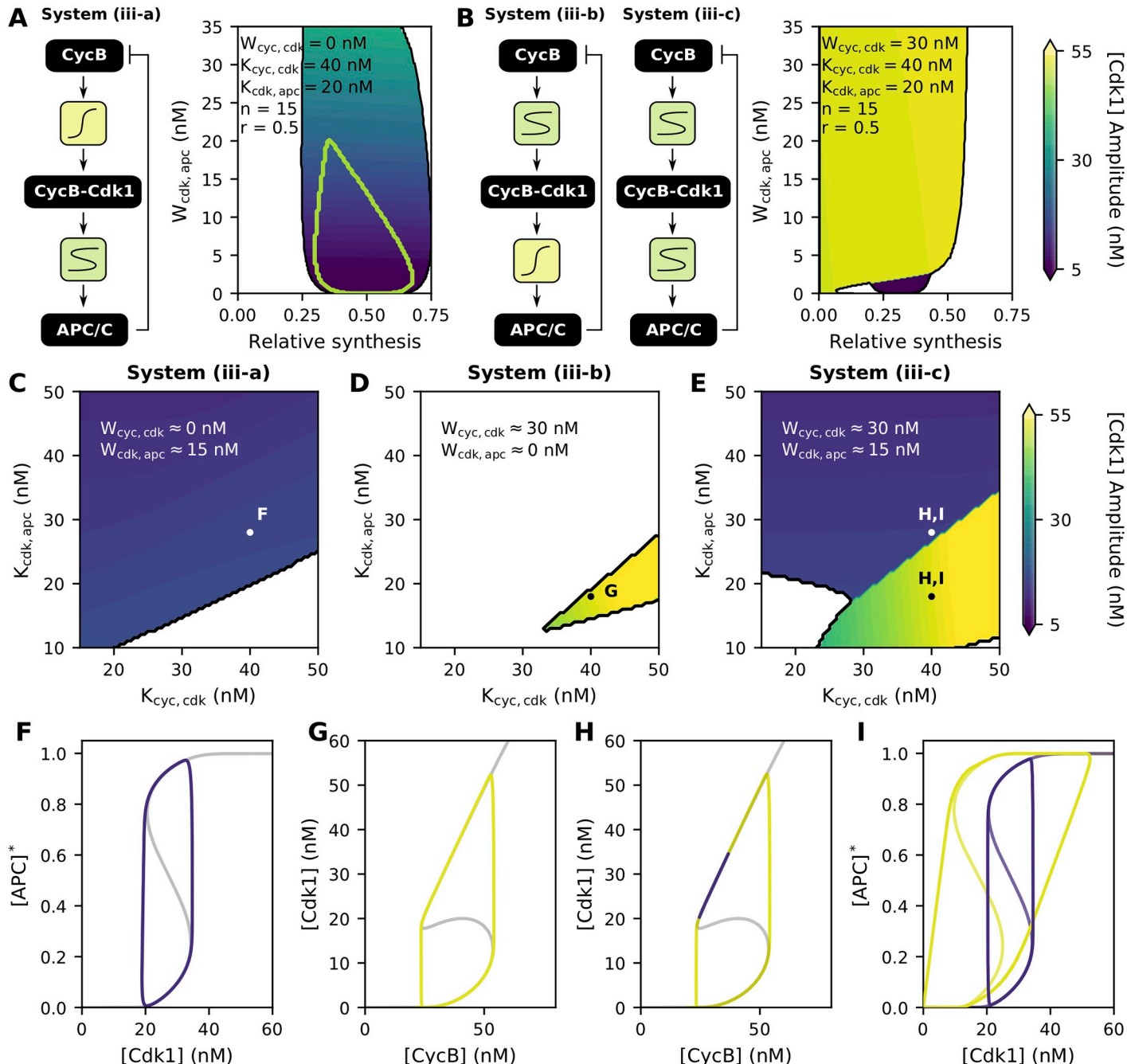

**Fig 6. Large amplitude oscillations are facilitated by two bistable switches.** Parameter values as indicated in the figure. **(A)** [Cdk1] amplitude for system (iii-a). The green contour represents the oscillatory region for system (i-b). **(B)** [Cdk1] amplitude for system (iii-b) and (iii-c). System (iii-b) corresponds to a horizontal cross-section at $W_{cdk,apc} = 0$. **(C-D)** [Cdk1] amplitudes for the different system configurations as a function of the threshold values $K$. Parameters as in panel B or as indicated, with $c = 0.5$. **(F-I)** Phase plane representation showing the relationship between the oscillations of the system and its steady-state response. Parameter values and colors (yellow vs purple) correspond to panels C-E.

compromise the applicability of the approach. In fact, it is required to obtain the correct shape of the response curve as experimentally determined [11].

Adding this second module for CycB-Cdk1 regulation greatly affects the position and shape of the nullclines, which now become surfaces in the three dimensional space (although they

can be projected in a 2D plane). As a result, the parameter space sustaining oscillations changes compared with system (i-b), even if the added response of [Cdk1] as a function of [CycB] is only ultrasensitive, as in system (iii-a) (Fig 6A, note how this latter system can be derived from system (iii) by setting the prefactor $\alpha_{cdk}$ in $\xi(^{[Cdk1]}/_{[CycB]})$ equal to zero). In particular, oscillations can be observed for larger widths of the S-shaped response curve. Whenever the [Cdk1] response as a function of [CycB] becomes S-shaped by making $\alpha_{cdk}$ in system (iii) greater than zero, the behavior of the system drastically changes, with two distinct types of oscillations emerging, corresponding to either low or high [Cdk1] amplitudes (purple vs yellow regions in Fig 6B respectively; system (iii-b) for $W_{cdk,apc} = 0$ and (iii-c) for $W_{cdk,apc} > 0$). These distinct types of behavior are not solely determined by the width of the response and the relative synthesis $c$ (as in Fig 6B), but also by the relative position of both nullclines. Indeed, whereas system (iii-a) and (iii-b) show either low or high amplitude [Cdk1] oscillations, independent of the threshold values $K$ (Fig 6C and 6D), small changes of these thresholds in system (iii-c) can entail a prompt transition between low and high amplitude oscillations (Fig 6E).

To clearly visualize the different types of oscillations just discussed, consider the phase-plane representations in Fig 6F, 6G, 6H and 6I. For system (iii-a) (Fig 6F), the time trajectory closely follows the [APC]* nullcline in the [Cdk1]-[APC]* plane, similar as was the case for system (i-b), resulting in relatively small [Cdk1] amplitudes. For system (iii-b) on the other hand (Fig 6G), the oscillations are dominated by the [Cdk1] nullcline, as seen from the projection in the [CycB]-[Cdk1] plane. As the [Cdk1] nullcline keeps rising indefinitely, large [Cdk1] amplitudes are established. For system (iii-c), both the [Cdk1]-[APC]* and [CycB]-[Cdk1] plane are shown (Fig 6H and 6I). It can now be appreciated that, depending on the relative position of the nullclines, the oscillations are either dominated by the [APC]* nullcline (i.e. small amplitude oscillations in purple) or by the [Cdk1] nullcline (i.e. large amplitude oscillations in yellow).

From a biological perspective, the high amplitude oscillations generated by two bistable switches induce a sufficiently large increase in CycB-Cdk1 activity followed by sufficiently large inactivation required for correct entry into and exit from mitosis. Furthermore, the two bistable switches allow the occurrence of such oscillations for a larger set of parameter values (compare the larger yellow region in Fig 6E with that in Fig 6D). This means that cell cycle oscillations can be ensured even if physiological conditions within the cell (e.g. enzyme activity) would fluctuate, a situation which might otherwise disrupt correct cell cycle progression.

## The cell cycle can be represented as a chain of interlinked bistable switches

The results on the G2-M transition coupled to a (delayed) negative feedback in the previous sections are mainly applicable to the early embryonic cell cycles that rapidly cycle between S phase and M phase without intermittent gap phases. Many insects, amphibians and fish that lay their eggs externally carry out multiple rounds of such rapid cell cycles following fertilization. However, all of them then pass through the so-called midblastula transition (MBT) after approximately ten cell cycles [64–67]. This transition is characterized by the establishment of gap phases and slowing down of the cell cycle, resulting in a higher resemblance to the cell cycle as typically studied in yeast and mammalian somatic cells. Remarkably, many of the transitions between these additional cell cycle phases are—just like the G2-M transition—governed by bistable switches [26].

During the G1-S transition in cultured mammalian cells for example, it is the activity of the E2F transcription factor that is regulated in a bistable manner. The external presence of growth factors can induce the expression of CycD and the activation of CycD-Cdk4/6 complexes, followed by the activation of E2F. E2F then induces the expression of several target genes, one of

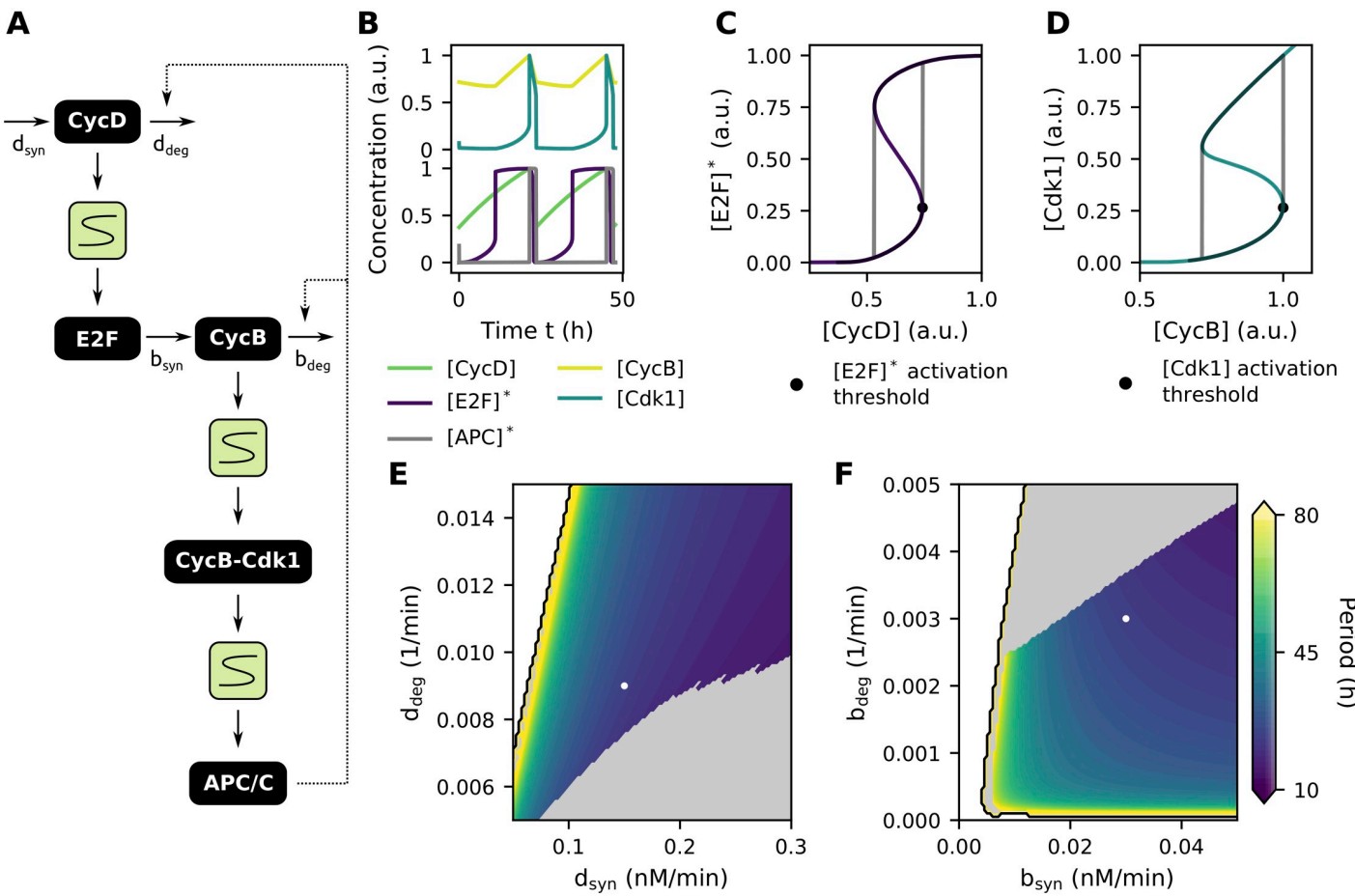

**Fig 7. The cell cycle can be represented as a chain of interlinked bistable switches.** Parameter values from Table 2 or as indicated in the figure. **(A)** Block scheme of interlinked S-shaped modules representing the overall cell cycle. **(B)** Oscillating time traces of the cell cycle model. **(C,D)** Time traces from panel B orbiting around the bistable response curves. As in panel B, concentrations are normalized for their maximal value. **(E,F)** Overall period of the oscillations as a function of CycD (E) or CycB (F) synthesis and degradation rates. No oscillations exist in the white regions. Grey regions represent irregular oscillations. White dots denote parameter values used in panels B-D.

which is CycE. As CycE can further enhance the activation of E2F, a positive feedback loop is established, ultimately leading to the bistable response of E2F [19]. Consequently, the G1-S transition can be depicted as an S-shaped module, similar as was done for the G2-M transition.

It is possible to combine the different transitions—each represented by an S-shaped module—into a phenomenological model of the overall cell cycle. For this, we still need to link the 'input' and 'output' of each module in a suitable way. We link E2F activity (the output of the G1-S switch) to CycB (the input of the G2-M switch) by recognizing that CycA, which is a target of E2F, drives the activation of the transcription factor FoxM1, which subsequently induces the expression of CycB [68] (Fig 7A, see Materials and methods for associated equations). Similar as before, cyclins then need to be degraded to reset the system to the lower branch of the S-shaped response curves, after which a new round of cell division can start. Here, it is assumed that the degradation of all cyclins is induced in M phase by activated APC/C.

As seen from Fig 7B, this modular representation of the cell cycle can produce oscillations in the concentration levels of the biochemical components. Synthesis of [CycD] (here used to directly model the effect of external growth factors), results in a linear increase of its

concentration levels. Once [CycD] levels reach the [E2F]$^*$ activation threshold, [E2F]$^*$ activity suddenly rises (Fig 7B and 7C). Activated [E2F]$^*$ then (indirectly) induces the expression of [CycB], followed by the activation of [Cdk1] when [CycB] levels cross the [Cdk1] activation threshold (Fig 7B and 7D). As in the previous model of the G2-M transition, active CycB-Cdk1 induces APC/C activation and subsequent degradation of the cyclins (Fig 7C and 7D, S1 Video).

The oscillatory behavior of this system depends on the synthesis and degradation rates of the cyclins, together with the shape of the response curves themselves. For certain combinations of parameters, no (white regions in Fig 7E and 7F) or irregular (grey regions in Fig 7E and 7F) oscillations are observed. With irregular oscillations, we mean that the system might get stuck on either the upper or lower branch of one of the S-shaped response curves, either transiently or permanently (see also S6 Fig). The duration of the different phases depends on the values of production and degradation rates. A smaller synthesis rate $d_{syn}$ of [CycD] and larger degradation rate $d_{deg}$, entails a longer time needed for [CycD] levels to reach the threshold value for [E2F]$^*$ activation. This leads to oscillations with elongated G1 phases and thus longer overall periods (Fig 7E and S7 Fig). A decrease in the synthesis rate $b_{syn}$ or degradation rate $b_{deg}$ of [CycB] brings about oscillations with increased periods (Fig 7F). Whereas the effect of the synthesis rate can mainly be attributed to an elongation of the S-G2 phase, a diminished degradation rate causes an extension of the M phase (S7 Fig).

The modular approach adopted here allows for easy extension of the model by additional bistable switches. In S8 Fig for example, we incorporate a switch of FoxM1 activity with respect to CycA levels, which has been proposed based on similarities with the E2F reaction network [69]. However, as the existence of this switch has not been experimentally validated, we do not include it in our further analysis.

## Control mechanisms affecting the bistable switches shape the dynamics of the cell cycle

Unlike the early embryonic cell cycle, the somatic cell cycle is no 'clock-like' oscillator with a fixed period, but instead a 'domino-like' oscillator in which tight control mechanisms or 'checkpoints' have been established that safeguard correct DNA replication and cell division. Hence, the cell cycle representation from the previous section—in which no such checkpoints were considered—needs to be refined. Interestingly, the bistable nature of the cell cycle transitions itself can provide the means for this regulation by shifting the (in)activation thresholds (i.e. folding points) of the S-shaped response curves to lower or higher levels [13, 70, 71].

In Fig 7 and S7 Fig, we showed how a change in the synthesis or degradation rate of cyclins entails a change in the duration of a specific cell cycle phase, and thus the duration of the overall cell cycle. Although these findings are in line with experimental results in [72], others have reported how alterations in the duration of one phase can be compensated by an opposite change in other phases, rendering the overall cell cycle period unchanged [73]. Even without considering molecular details, the phenomenological model allows the description of such behavior. A decline in the overall cell cycle period due to enhanced [CycD] synthesis for example (Fig 8A,1–2), can be compensated by widening the S-shaped response of the subsequent phase (Fig 8A,2–3).

Next, let us consider the so-called restriction point (RP). The traditional view of this checkpoint states that cells in G1 phase are predetermined to stay in this phase and not enter S phase. Only if sufficiently high concentrations of growth factors are present in the extracellular environment, leading to sufficiently high synthesis rates of CycD, the cell will progress into S phase and irreversibly commit to complete the started round of cell division [74]. Interlinked

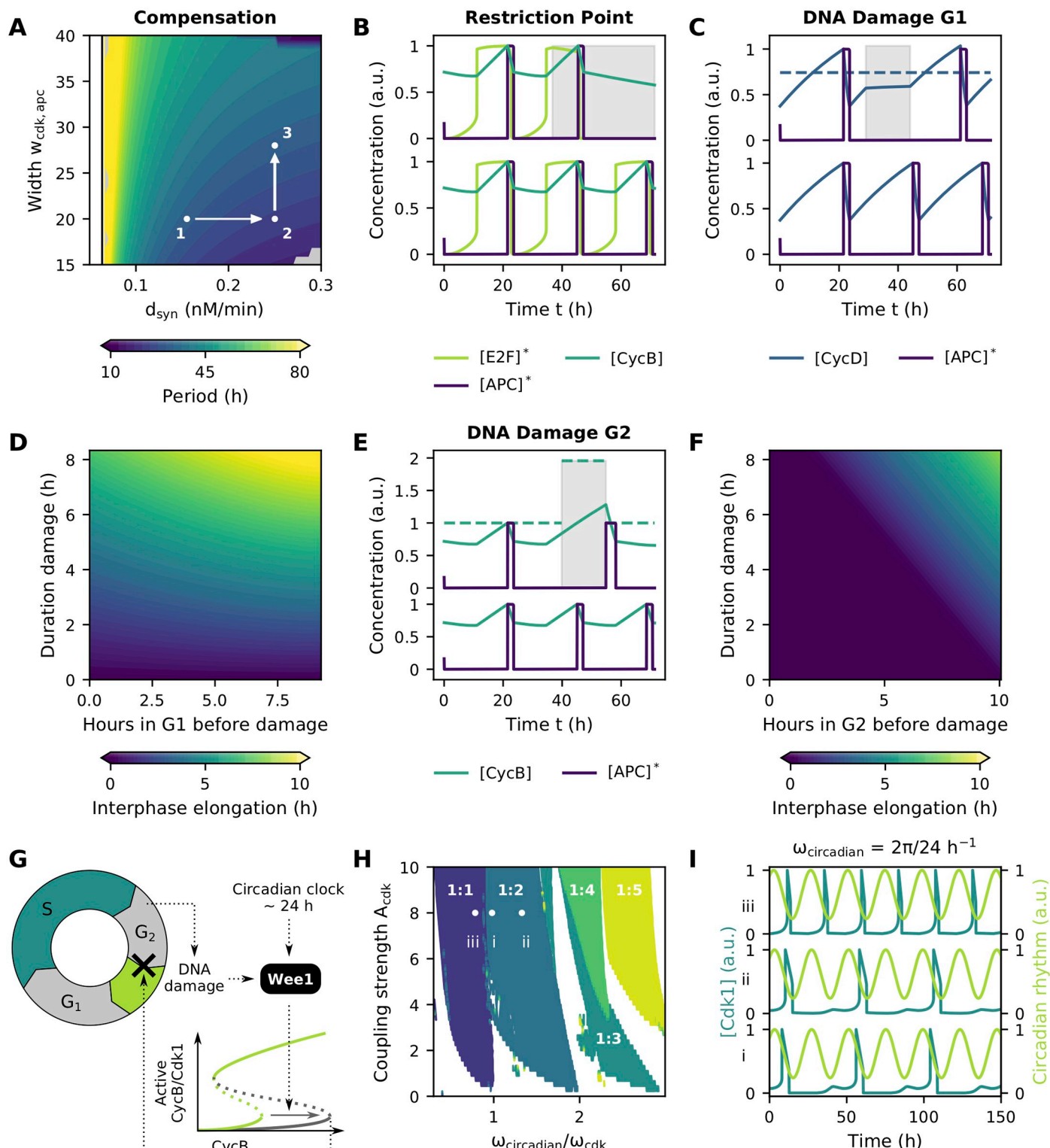

**Fig 8. Control mechanisms affecting the bistable switches shape the dynamics of the cell cycle.** Parameter values as in Table 2 or as indicated. **(A)** Potential changes in the overall cell cycle period, for example due to increased [CycD] synthesis (1–2), can be compensated by changes in subsequent phases, such as altering the width of the S-shaped responses (2–3). **(B)** Time traces comparing unperturbed cell cycle progression (bottom) with progression after reduction of the [CycD] synthesis rate. In line with the classical interpretation of the RP, a started round of cell division can still be completed (as seen from the activation of [APC]*) even if [CycD] synthesis is reduced (grey area). **(C)** Time traces comparing unperturbed cell cycle progression (bottom) with progression after DNA damage in G1 phase.

Dotted line represents the [E2F]* activation threshold and grey area marks region of increased [CycD] degradation. **(D)** Elongation of interphase after DNA damage in G1. **(E)** Time traces comparing unperturbed cell cycle progression (bottom) with progression after DNA damage in G2 phase. Dotted line represents the [Cdk1] activation threshold. Similar as in A and B, concentrations are normalized for their maximal value in the unperturbed case. **(F)** Elongation of interphase after DNA damage in G2. **(G)** The kinase Wee1 is controlled by DNA damage and circadian rhythms and affects the Cdk1 activation threshold. **(H)** Regions of p:q phase locking between [Cdk1] oscillations and the circadian clock. The coupling strength $A_{cdk}$ represents the amplitude of the sine wave used to model the periodically changing value of $\alpha_{cdk}$. The annotations i, ii, and iii refer to panel I. **(I)** Time traces of phase locking for constant circadian frequency (i.e. $\frac{2\pi}{24}$ h$^{-1}$) but different natural frequencies of the cell cycle ((i) $\omega_{cdk} \approx \frac{2\pi}{24}$ h$^{-1}$, (ii) $\omega_{cdk} \approx \frac{2\pi}{32}$ h$^{-1}$, (iii) $\omega_{cdk} \approx \frac{2\pi}{19}$ h$^{-1}$).

bistable switches can account for such behavior as—once activated—[E2F]* activity can remain high even if [CycD] synthesis is strongly reduced (grey area in Fig 8B). This high [E2F]* activity then further induces [CycB] expression to reach the [Cdk1] activation threshold followed by activation of [APC]*.

Another example of cellular control is the way cells preserve genomic integrity by arresting the cell cycle whenever deleterious DNA lesions are encountered. During G1 phase, DNA damage can prevent further cell cycle progression by inducing the degradation of CycD [75]. This behavior can be understood in our model by looking at the bistable switch governing the G1-S transition. The increased degradation of [CycD] (grey area in Fig 8C) keeps its levels below the [E2F]* activation threshold, thus blocking S phase entry, until DNA repair re-establishes normal degradation rates and [CycD] levels can rise beyond the threshold. The extent to which interphase is lengthened depends on when in G1 the damage was imposed, the duration of the damage and the time required to reach the [E2F]* activation threshold after the damage was repaired (Fig 8C and 8D).

DNA repair mechanisms during G2 phase have been associated with the activation of the kinase Wee1, which plays an important role in inactivating CycB-Cdk1 complexes and thus preventing entry into M phase (Fig 8G) [76]. Exploiting the fact that increased Wee1 activity shifts the Cdk1 activation threshold to higher CycB levels [13, 59], the effect of DNA damage can straightforwardly be implemented by tuning the $\alpha$ parameter of the S-shaped module. As long as DNA damage is present, the [Cdk1] activation threshold is shifted to the right (grey area in Fig 8E) and [CycB] levels keep rising until a steady state is reached. When extensive damage is encountered, a permanent raise of the threshold might result in permanent cell cycle arrest. However, if DNA damage is repaired, Wee1 gets inhibited again, the activation threshold shifts back to lower [CycB] levels and [Cdk1] is activated, resulting in subsequent [APC]* activation. From the time traces (Fig 8E), it can be observed that cells enter mitosis with higher [CycB] levels (and thus also higher [Cdk1] levels) when recovering from DNA damage in comparison with unperturbed cells. Consequently, a longer time period is required to reach the [APC]* inactivation threshold and M phase is prolonged. In contrast with the DNA checkpoint in G1 phase, modification of the bistable response curve in G2 can keep the total duration of interphase unaffected after DNA damage. If the damage is repaired before [CycB] levels reach the original [Cdk1] activation threshold, shifting it to higher levels has no effect on interphase duration (Fig 8F).

These examples show that a modular description can be used to implement biological events such as DNA damage in a phenomenological way: by adjusting the overall effect on the shape of the S-shaped response, without needing to know the exact molecular mechanisms. We provide another example of this approach by linking the cell cycle to the 24h circadian clock. Such coupling between the circadian clock and cell cycle has been studied extensively, both experimentally as well as theoretically (e.g. [77, 78]). Here, we use the fact that the expression of the Wee1 kinase is not only affected by DNA damage but also tightly regulated by the Clock-Bmal1 transcription factor complex, a master regulator of the circadian clock (Fig 8G) [79]. We explored the dynamics of such unidirectional coupling by

introducing the circadian regulation of Wee1 into the model. More specifically, the [Cdk1] activation threshold—which is regulated by Wee1—was periodically shifted between its basal [CycB] level and a predefined maximal level by explicitly modeling parameter $\alpha_{\text{cdk}}$ as a sine wave (see Materials and methods for details and ODEs). When two oscillators are coupled like this, one can expect to observe so-called p:q phase locking, meaning that p cycles of the cell cycle are completed for q cycles of the circadian clock. Whether such behavior indeed occurs depends on two factors [80]: (1) the relative frequencies of the circadian clock and the unforced cell cycle (i.e. the frequency in the absence of any circadian control, also called the natural frequency), and (2) on the strength of the coupling (here $A_{\text{cdk}}$, i.e. the amplitude of the sine wave used to model $\alpha_{\text{cdk}}$).

A commonly used approach to visualize phase-locking of two coupled oscillators, is plotting the regions where locking occurs as function of their frequency ratio and coupling strength. When represented like this, these regions of locking (referred to as Arnold tongues) are typically wedge-shaped (at least below a certain critical coupling strength) [80]: for low coupling strengths, p:q phase locking would only occur when the natural frequency of the cell cycle is close to the frequency of the circadian clock, whereas a larger mismatch between the two frequencies may still result in synchronization if the coupling strength increases. In our case, we do find these regions (see S9 Fig for time traces), but the Arnold tongues solely widen to the left hand side (Fig 8H). This indicates that the circadian clock only seems capable of lengthening the cell cycle and not shorten it. Indeed, examining the time traces in Fig 8I shows that a constant circadian rhythm of 24h elongates cell cycles with a natural period of 24h (i), 32h (ii) and 19h (iii) to forced periods of 48h, 48h and 24h, respectively. Considering that the circadian clock in our model only shifts the [Cdk1] activation threshold to higher [CycB] levels, relative to those of the unforced cell cycle, explains why the forced cell cycle period cannot become shorter than the unforced one. This observation is in agreement with the findings in [81], where unidirectional regulation of Wee1 by the circadian clock was analyzed using a mechanistic model. There too, Arnold tongues were found to only widen to the left, at least when a basal Wee1 synthesis rate was included. It should be emphasized that the unidirectional coupling between circadian clock and cell cycle as discussed here, merely functions as a conceptual example of our phenomenological modeling approach. Indeed, in reality, the situation is much more complex, as coupling happens in a bidirectional manner [77, 78] and also the circadian clock itself includes bistable switches [82]. Nevertheless, as with the other examples discussed, certain key aspects of the system's dynamical behavior can be described using the approach proposed here.

## Discussion

In an attempt to unravel the complexity of living systems, it is convenient to envision them as a hierarchical structure of interlinked biological modules. Under the premise that the functionalities of these individual modules do not change when combined with each other, knowing their behavior allows to predict the behavior of the overall system [35]. Although it has been recognized for a long time [83] that complex living systems cannot always be understood as the sum of their constituent parts, many examples of modularity do exist. Two proteins (or protein domains), for example, can be combined to generate recombinant proteins such as fused fluorescent reporters. Additionally, transcriptional promoters can be combined in synthetic networks that possess desired dynamical features, such as the oscillatory behavior of the 'repressilator', which consists of three negative feedback modules [84]. Furthermore, solutions have been proposed to overcome potential invalidity of the modular approach in engineered biological circuits [35]. Regarding the modules considered in this work, bistability itself has

been shown to allow for a modular approach [39, 40], whereas the validity of coupling ultra-sensitive modules has been described elsewhere [85].

To gain insight into the dynamics of individual or interlinked modules that are difficult to understand in an intuitive way, mathematical models can be very useful. As such, they also provide helpful tools to guide the design of synthetic biological systems [86]. One way to set up a mathematical model is to start from known biochemical interactions and use the law of mass action to derive kinetic rate equations for the concentrations of the biochemical components. However, such mechanistic models often contain a large number of variables and parameters, many of which can be difficult to measure experimentally. This can impede generalization of the obtained results, which might strongly depend on the particular set of chosen parameter values. Therefore, techniques for reducing the complexity of mathematical models while preserving their most fundamental characteristics have been a topic of great interest for several decades, ranging from the well-established quasi-steady-state assumption to more advanced mathematical algorithms [87–90].

A conceptually easy approach to reduce the complexity of biochemical equations is to replace the detailed reaction mechanisms of certain biological modules by a mathematical function that explicitly describes their resulting dynamical behavior. Such a functional or phenomenological approach not only reduces the complexity of mathematical models, but it can also be a valuable alternative to model biochemical networks for which the underlying interaction patterns have not been identified completely. Furthermore, parameters of functional modules correspond directly to observable measurements that are often easier to determine experimentally than the values of biochemical rate constants. In our example of a bistable switch, the parameter $\alpha$ correlates with the width of the bistable region. By choosing the function $\xi$ appropriately, the activation and inactivation threshold can thus be set directly in accordance with a response curve obtained in the lab. Moreover, the fact that these features are directly tunable allows us to answer questions such as 'how does the activation threshold influence the period of oscillations?'. In a mechanistic model, answering such questions would depend on knowledge about what biochemical parameters determine the activation threshold, which might be less straightforward. The same argument holds for other properties, such as steepness of a response or a time delay. Thus, a direct intuitive link between the model and experimentally determined response curves exists when using functional modules. This is an important advantage of this strategy over other frameworks that are suitable for modeling biochemical reactions without considering mechanistic details, such as Boolean networks [91], especially for biologists without mathematical background.

Functional modules based on the Hill function have been extensively used to describe steep sigmoidal response curves, which can originate from various biochemical mechanisms (e.g. multisite phosphorylations, cooperative binding events and stoichiometric inhibition) and the validity of which has been recently reviewed in [63]. All of the molecular details that may generate such a response are then hidden in a few parameters such as the exponent $n$ and a threshold $K$. In contrast, S-shaped response curves are typically not included explicitly in mathematical models. An important reason for this might be that an S-shaped response is not a function, because there are multiple output values for a single input. One contribution of this paper is to show that, in fact, a Hill function can be smoothly changed into an S-shaped response curve by making the threshold dependent on the output variable. In this way, the equation

$$\frac{dY}{dt} = \frac{X(t - \tau)^n}{(K\xi(Y))^n + X(t - \tau)^n} - Y \tag{3}$$

provides an easy way to model a system with input $X$ and output $Y$ that includes the three modules of ultrasensitive, S-shaped and time delayed responses. If the function $\xi$ is equal to 1, we recover an ultrasensitive response for the steady state of $Y$ as function of $X$. By choosing and tuning $\xi$, many different shapes of S-shaped response curves can be obtained. Whenever an experimentally determined steady-state curve is available, the function $\xi$ could also be fitted to such data (similar as what we did when fitting the S-shaped module to the mass-action model). An important condition for this phenomenological equation to correctly describe the dynamics of the overall system, is the separation of timescales so that the time traces closely follow the (ultrasensitive or S-shaped) steady-state response curves, at least in the absence of any time delay. In the case of time delays, validity criteria for modeling them in a phenomenological way remain largely absent, although it has been argued that the way a delay is implemented in a model should be chosen carefully [60].

In the first part of the paper, we used this extended Hill function in combination with a negative feedback loop to model the early embryonic cell cycle oscillator and showed how the shape of the steady-state response affects the oscillations. For ultrasensitive responses, no oscillations can be observed unless a sufficiently large time delay is present (S3 Fig, [60]). On the other hand, a system containing an S-shaped module readily sustains oscillations. Whether oscillations occur depends on the width of the S-region, and oscillations are more likely if there are time delays in the system. A second bistable switch can also facilitate oscillations and furthermore gives rise to two distinct 'oscillatory regimes', i.e. high vs low [Cdk1] amplitude oscillations. These findings are in line with previous work showing how ultrasensitivity, bistability and delay embedded in a negative feedback loop promote oscillatory behavior of a system [13, 46, 54]. Whereas most of this earlier work has studied these different modules in isolation, we provided a straightforward method to study their combined effect.

In the second part, we extended the phenomenological model of the early embryonic cell cycle with an additional bistable switch for the G1-S transition. Furthermore, these bistable switches were exploited to model several cell cycle checkpoints, thus making the model representative for the somatic cell cycle. Even if the molecular underpinnings of these checkpoints are not incorporated into the model, the functional approach can give some basic understanding of events such as the restriction point (RP), DNA damage checkpoints and even coupling of the cell cycle with the circadian clock.

The restriction point as depicted here corresponds to the classical interpretation where newborn cells have to cross a threshold in order to activate E2F and commit to S phase. It should be noted that such representation is a simplification, as it is known that additional bistable switches control entry into S phase [92–94]. Furthermore, some cells retain hyperphosphorylated Rb proteins, which are incapable of inhibiting E2F, after division. These cells can directly start S phase even in suboptimal conditions, challenging the classical view of the RP [74]. Interestingly, irregular cell cycle progression in which E2F activity remained high could also be observed for several combinations of parameters in our phenomenological model (S6 Fig). It would be interesting to see whether the biochemical mechanisms enabling cells to bypass the RP correspond with changes in functional responses identified here.

By implementing DNA damage in G2 via manipulation of the Cdk1 activation threshold, we found that cells enter M phase with higher CycB-Cdk1 levels after DNA damage in G2 phase. This is in line with recently published experimental results [95]. Whether the subsequent M phase takes longer than in unperturbed cells, as is predicted by the model, was not assessed in the cited study. If this would not be the case, it could indicate that either

CycB degradation is accelerated after DNA damage or that the inactivation threshold too is shifted to higher CycB levels. Another interesting study which looked at the effects of DNA repair mechanisms was published by Chao *et al.*, [96]. There, the authors found that elongation of interphase after DNA damage in G2 depends on the severity of the damage, but is independent of when the cells were damaged within G2. This finding is in contrast with our model predictions that DNA damage in G2 can actually be repaired without interphase elongation. It should be noted that here we supposed that DNA damage can be detected and repaired at any time during G2, not only at a checkpoint at the end of G2 phase. Furthermore, our model does not account for regulatory processes such as decreased CycB synthesis [97] or increased CycB degradation [98], which might explain the discrepancies. Even if we cannot account for molecular details and nuances, the possibility to incorporate DNA damage in a phenomenological manner can be helpful to understand these events in a qualitative way.

The early embryonic cell cycle can be considered as a 'clock-like' oscillator, which from a dynamical systems viewpoint consists of an autonomous limit cycle oscillator [3, 99]. For more complex cell cycles, on the other hand, such as the mammalian cycle with its checkpoints, the dynamical nature is not entirely clear yet. Some authors suggest that the latter too, is a limit cycle in which the ordering of different cell cycle phases and the activity of different Cdk's naturally emerge from the underlying biochemical interaction network [100]. Another viewpoint regards the cell cycle as a series of irreversible transitions, each one generated by a bistable switch. In that case, changes in the—what we called—input variable of the switches drive the system forward and a cell cycle transition happens whenever a certain input threshold is reached. The input variable can be several things, such as mitogen concentrations, cyclin levels or cell mass [55]. Whether autonomous limit cycle oscillations can or cannot be observed, then depends on whether the dynamics of the driving force is or is not implemented in the system equations, respectively.

Our model contains elements from both the viewpoint of the cell cycle as a limit cycle and as a chain of bistable switches. The equations we used describe changes in cyclin levels, which drive the cell through the different transitions. The cell cycle emerges as an autonomous limit cycle with a fixed period. It is, however, a complicated limit cycle that conceptually differs from a simple oscillator, as the trajectory jumps between distinct states (corresponding to cell cycle phases). Moreover, our model allows for the implementation of checkpoints, i.e. temporarily pausing the spontaneous oscillations, by altering the rate constants or the underlying S-shaped responses themselves. Granted, this checkpoint action is then generated by an external input or perturbation, which makes the cycle non-autonomous. As such, our model shows features of both domino-like and clock-like behavior, and is similar in spirit to, for example, the model studied by Gonze and Goldbeter [101].

Although in this work we combined the S-shaped/ultrasensitive/delay modules to generate a phenomenological model of the overall cell cycle, these modules are also applicable for modeling other processes based on their experimentally determined response curves. Bistable switches for example, are known to play a role during epithelial-to-mesenchymal transition [102] and differentiation in cell types ranging from embryonic stem cells [103] to osteogenic precursors [104] and *Drosophila* eyes [105]. Furthermore, the convenient way of combining several functional modules can be exploited to reconstitute interaction patterns with complex dynamical behavior, such as a combination of the 'repressilator' with bistable switches that has been observed during neural tube development [106].

The modeling strategy based on functional modules put forth in this work fits in the mindset of reducing the complexity of mathematical models and has the advantage of preserving an

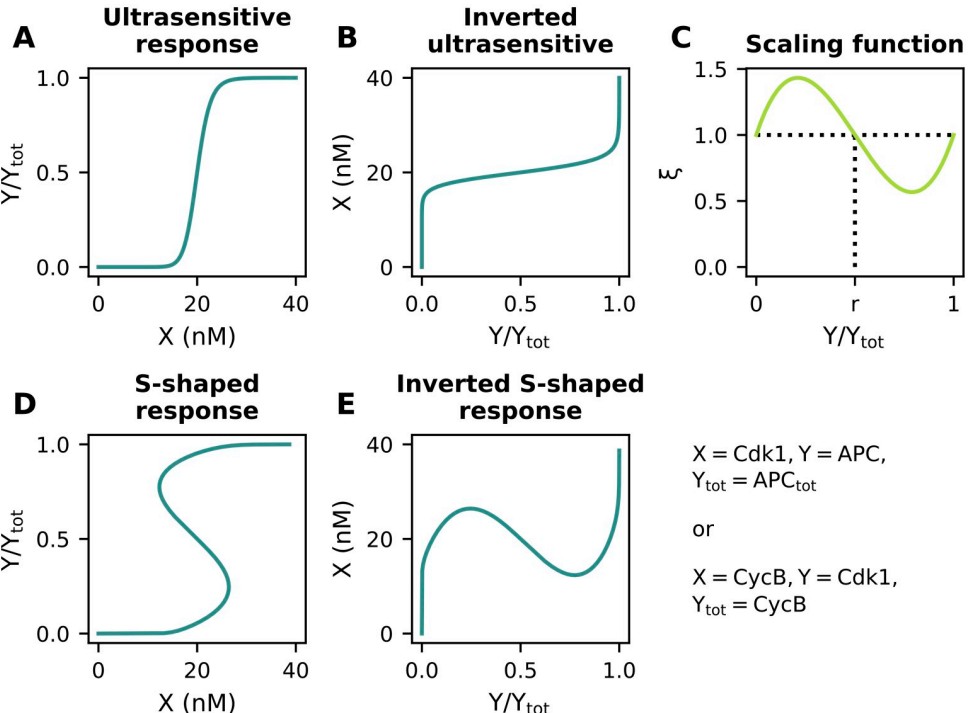

**Fig 9.** An ultrasensitive response curve of variable $Y$ normalized for its total concentration $Y_{tot}$ (A) can be converted into an S-shaped one (D) by inverting the product (E) of a scaling function (C) with the inverted ultrasensitive response (B).

intuitive link with the experimentally measured response curves. Furthermore, the modular approach provides the flexibility for combining the modules in a multitude of different ways. As such, it can provide a toolbox for other researchers who want to generate phenomenological modular models, not only of the cell cycle but also of many other biological processes.

## Materials and methods

### Converting an ultrasensitive function into an S-shaped one

Starting from an (increasing) ultrasensitive response curve, one can obtain an S-shaped response by shifting the lower and upper part of the curve to the right and left respectively. For APC/C, the ultrasensitive response curve can be expressed as a Hill function representing the fraction of activated APC/C to total APC/C molecules as a function of [Cdk1] (Fig 9A):

$$[APC]^* = \frac{[APC/C]}{[APC/C]_{tot}} = \frac{[Cdk1]^n}{K_{cdk,apc}^n + [Cdk1]^n} \tag{4}$$

The S-shaped $[APC]^*$ response cannot be expressed as a function of [Cdk1] (since it would have multiple $[APC]^*$ values for a certain range of [Cdk1]). However, even if the function is S-shaped, there would be only one [Cdk1] value for each $[APC]^*$ value. This motivates us to

invert the function (Fig 9B):

$$[\text{Cdk1}] = K_{\text{cdk,apc}} \left( \frac{[\text{APC}]^*}{1 - [\text{APC}]^*} \right)^{1/n}$$

Subsequently, this inverted response can be multiplied by the scaling function $\xi([\text{APC}]^*)$ (Fig 9C), such that [Cdk1] values increase for low $[\text{APC}]^*$ values and decrease for high $[\text{APC}]^*$ (Fig 9E):

$$[\text{Cdk1}] = \xi([\text{APC}]^*) \cdot K_{\text{cdk,apc}} \left( \frac{[\text{APC}]^*}{1 - [\text{APC}]^*} \right)^{1/n} \tag{5}$$

with $\xi([\text{APC}]^*) = 1 + \alpha_{\text{apc}} \cdot [\text{APC}]^* ([\text{APC}]^* - 1) ([\text{APC}]^* - r)$. Rearranging Eq 5 leads to an expression for an S-shaped response (Fig 9D), analogous with Eq 1 in the main text. This is not a function, but the expression can be put in a differential equation such that the steady state of this equation follows the S-shaped response:

$$[\text{APC}]^* = \frac{[\text{Cdk1}]^n}{(\xi([\text{APC}]^*) \cdot K_{\text{cdk,apc}})^n + [\text{Cdk1}]^n}$$

$$\Rightarrow \frac{d[\text{APC}]^*}{dt} = \frac{1}{\epsilon_{\text{apc}}} \left( \frac{[\text{Cdk1}]^n}{(\xi([\text{APC}]^*) \cdot K_{\text{cdk,apc}})^n + [\text{Cdk1}]^n} - [\text{APC}]^* \right) \tag{6}$$

A similar approach can be followed for the bistable switch from [CycB] to [Cdk1]. The ultrasensitive response is given by

$$\frac{[\text{Cdk1}]}{[\text{CycB}]} = \frac{[\text{CycB}]^n}{K_{\text{cyc,cdk}}^n + [\text{CycB}]^n} \tag{7}$$

This is similar to Eq 4. Indeed the amount of active CycB-Cdk1 complexes is normalized for the total amount of CycB-Cdk1 complexes, which is given by the amount of CycB molecules because Cdk1 is present in excess and it is assumed that free CycB molecules immediately bind free Cdk1 molecules [107]. Rearranging and multiplying by $\xi(^{[\text{Cdk1}]}/_{[\text{CycB}]}) = 1 + \alpha_{\text{cdk}} \cdot$ $^{[\text{Cdk1}]}/_{[\text{CycB}]}(^{[\text{Cdk1}]}/_{[\text{CycB}]} - 1)(^{[\text{Cdk1}]}/_{[\text{CycB}]} - r)$ gives:

$$[\text{CycB}] = \xi\left(^{[\text{Cdk1}]}/_{[\text{CycB}]}\right) \cdot K_{\text{cyc,cdk}} \left( \frac{^{[\text{Cdk1}]}/_{[\text{CycB}]}}{1 - ^{[\text{Cdk1}]}/_{[\text{CycB}]}} \right)^{1/n}$$

$$\Rightarrow \frac{[\text{Cdk1}]}{[\text{CycB}]} = \frac{[\text{CycB}]^n}{\left( \xi\left(^{[\text{Cdk1}]}/_{[\text{CycB}]}\right) \cdot K_{\text{cyc,cdk}} \right)^n + [\text{CycB}]^n} \tag{8}$$

This means that

$$[\text{Cdk1}] = \frac{[\text{CycB}]^n}{\left( \xi\left(^{[\text{Cdk1}]}/_{[\text{CycB}]}\right) \cdot K_{\text{cyc,cdk}} \right)^n + [\text{CycB}]^n} [\text{CycB}] \tag{9}$$

Again, this is not a function, but can be made the steady state of a differential equation (system (iii) in the main text):

$$\frac{d[\text{Cdk1}]}{dt} = \frac{1}{\epsilon_{\text{cdk}}}\left(\frac{[\text{CycB}]^n}{\left(\xi\left([\text{Cdk1}]/_{[\text{CycB}]}\right) \cdot K_{\text{cyc,cdk}}\right)^n + [\text{CycB}]^n}[\text{CycB}] - [\text{Cdk1}]\right)$$

It needs to be emphasized that both Eqs 4 and 7 describe an ultrasensitive response of which the output is given by the concentration level of a compound $Y$, normalized for the total amount $Y_{\text{tot}}$. Accordingly, both expressions are limited to the interval $Y/_{Y_{\text{tot}}} \in [0, 1]$ (Fig 9A). Similarly, Eqs 6 and 8 describe S-shaped responses for $Y/_{Y_{\text{tot}}} \in [0, 1]$ (Fig 9D). When incorporating these steady-state expressions into the differential equations however, a distinction should be made. As $[\text{APC/C}]_{\text{tot}}$ is a constant, the ratio $[\text{APC}]^* = {}^{[\text{APC/C}]}/_{[\text{APC/C}]_{\text{tot}}}$ can directly be incorporated into the ODE for the rate of change of $[\text{APC}]^*$. In contrast, $[\text{CycB}]$ is not a constant and therefore, the rate of change of $^{[\text{Cdk1}]}/_{[\text{CycB}]}$ would need to be given by the quotient rule for differentiation, as both numerator and denominator are variables. Alternatively, $[\text{CycB}]$ can first be moved to the right hand side of Eq 8, resulting in Eq 9. Subsequently, this expression for the steady state of $[\text{Cdk1}]$ can be incorporated into an ODE describing the rate of change of $[\text{Cdk1}]$. Of note, Eq 9 does no longer saturate at $[\text{Cdk1}] = 1$ (as was the case for Eq 8 at $^{[\text{Cdk1}]}/_{[\text{CycB}]} = 1$) but keeps rising indefinitely (see for example Fig 6G and 6H).

## Non-dimensionalization of system equations

To facilitate mathematical analysis, the system equations for the two-variable models described in the main text were non-dimensionalized by introducing the new variables:

$$[\text{Cdk1}]^* = \frac{[\text{Cdk1}]}{K_{\text{cdk,apc}}} \qquad [\text{APC}]^* = \frac{[\text{APC/C}]}{[\text{APC/C}]_{\text{tot}}} \qquad t^* = b_{\text{deg}} \cdot t \qquad \tau^* = b_{\text{deg}} \cdot \tau$$

where $b_{\text{deg}}$ [min$^{-1}$] is the reaction rate of the first order degradation of CycB-Cdk1 complexes (i.e. variable $[\text{Cdk1}]$) by $[\text{APC}]^*$. As an example, system (ii) then becomes:

$$\begin{cases} \dfrac{d[\text{Cdk1}]^*}{dt^*} = \dfrac{b_{\text{syn}}}{b_{\text{deg}} \cdot K_{\text{cdk,apc}}} - [\text{Cdk1}]^* \cdot [\text{APC}]^* \\[2ex] \dfrac{d[\text{APC}]^*}{dt^*} = \dfrac{1}{\epsilon_{\text{apc}} \cdot b_{\text{deg}}}\left(\dfrac{[\text{Cdk1}]^{*n}(t^* - \tau^*)}{\xi([\text{APC}]^*)^n + [\text{Cdk1}]^{*n}(t^* - \tau^*)} - [\text{APC}]^*\right) \end{cases} \qquad (10)$$

after which we can define $c = \frac{b_{\text{syn}}}{b_{\text{deg}} \cdot K_{\text{cdk,apc}}}$, $\epsilon_{\text{apc}}^* = \epsilon_{\text{apc}} \cdot b_{\text{deg}}$ and $\xi([\text{APC}]^*)$ remains unaffected:

$$\xi([\text{APC}]^*) = 1 + \alpha_{\text{apc}} \cdot [\text{APC}]^*([\text{APC}]^* - 1)([\text{APC}]^* - r)$$

The positive production term in the ODE for $[\text{Cdk1}]$ is referred to as the 'relative synthesis rate' and here equals $c$. Standard parameter values used are: $\epsilon_{\text{apc}}^* = 0.01$, $n = 15$, $b_{\text{deg}} = 0.1$ min$^{-1}$, $K_{\text{cdk,apc}} = 20$ nM, of which the latter three are based on experimental observations in *Xenopus laevis* eggs [15, 30]. Obtained simulation results were scaled back to dimensional values where possible. For APC/C however, no experimental estimates for the total amount $[\text{APC/C}]_{\text{tot}}$ were found and results are presented as $[\text{APC}]^*$.

The system containing two S-shaped response (system (iii)) was non-dimensionalized in a similar way:

$$[\text{CycB}]^* = \frac{[\text{CycB}]}{K_{\text{cyc,cdk}}} \qquad [\text{Cdk1}]^* = \frac{[\text{Cdk1}]}{K_{\text{cdk,apc}}} \qquad [\text{APC}]^* = \frac{[\text{APC/C}]}{[\text{APC/C}]_{\text{tot}}} \qquad t^* = b_{\text{deg}} \cdot t$$

$$\begin{cases} \dfrac{d[\text{CycB}]^*}{dt^*} = \dfrac{b_{\text{syn}}}{b_{\text{deg}} \cdot K_{\text{cyc,cdk}}} - [\text{CycB}]^* \cdot [\text{APC}]^* \\[3mm] \dfrac{d[\text{Cdk1}]^*}{dt^*} = \dfrac{1}{\epsilon_{\text{cdk}} \cdot b_{\text{deg}}} \left( \dfrac{K_{\text{cyc,cdk}}}{K_{\text{cdk,apc}}} \cdot \dfrac{[\text{CycB}]^{*n+1}}{\xi\left({[\text{Cdk1}]^*}/{d \cdot [\text{CycB}]^*}\right)^n + [\text{CycB}]^{*n}} - [\text{Cdk1}]^* \right) \\[3mm] \dfrac{d[\text{APC}]^*}{dt^*} = \dfrac{1}{\epsilon_{\text{apc}} \cdot b_{\text{deg}}} \left( \dfrac{[\text{Cdk1}]^{*n}}{\xi([\text{APC}]^*)^n + [\text{Cdk1}]^{*n}} - [\text{APC}]^* \right) \end{cases} \qquad (11)$$

If $\frac{K_{\text{cyc,cdk}}}{K_{\text{cdk,apc}}} = d$, we get $\frac{b_{\text{syn}}}{b_{\text{deg}} \cdot K_{\text{cyc,cdk}}} = \frac{c}{d}$. For the scaling functions we have $\xi([\text{APC}]^*)$ as before and

$$\xi\left({[\text{Cdk1}]^*}/{d \cdot [\text{CycB}]^*}\right) = 1 + \alpha_{\text{cdk}} \cdot \frac{[\text{Cdk1}]^*}{d \cdot [\text{CycB}]^*} \left( \frac{[\text{Cdk1}]^*}{d \cdot [\text{CycB}]^*} - 1 \right) \left( \frac{[\text{Cdk1}]^*}{d \cdot [\text{CycB}]^*} - r \right)$$

Note how the relative synthesis rate (again the positive production term of cyclins) is given by $c/d$ for this system, rather than $c$. The same standard parameter values as the two-dimensional system were used, with the additional parameters being: $b_{\text{deg}} \epsilon_{\text{cdk}} = b_{\text{deg}} \epsilon_{\text{apc}} = 0.01$ and $K_{\text{cyc,cdk}} = 40$ nM [11].

## Conversion of parameter $\alpha$ to the width of the S-shaped region

All screens for which the width of the bistable region was altered, were performed by screening different values of $\alpha$ and linking this value to the width of the S-shaped region. The expression for the non-dimensionalized S-shaped response curve, i.e.

$$y = \frac{x^n}{\xi(y)^n + x^n} \qquad \text{with} \begin{cases} x = [\text{Cdk1}]^*, y = [\text{APC}]^* \\[2mm] x = [\text{CycB}]^*, y = {[\text{Cdk1}]^*}/{d \cdot [\text{CycB}]^*} \end{cases},$$

can be inverted, resulting in:

$$x = \xi(y) \left( \frac{y}{1-y} \right)^{1/n} \qquad \text{with} \begin{cases} x = [\text{Cdk1}]^*, y = [\text{APC}]^* \\[2mm] x = [\text{CycB}]^*, y = {[\text{Cdk1}]^*}/{d \cdot [\text{CycB}]^*} \end{cases}$$

The width of the S-shaped region is given by the difference of $x$-values at the local extrema of this inverted S-shaped response, which can be calculated as the roots of the derivative. For the

cubic scaling function $\xi(y) = 1 + \alpha[y^3 - (1+r)y^2 + r \cdot y]$, we have:

$$\frac{dx}{dy} = \frac{d\xi}{dy} \cdot \left(\frac{y}{1-y}\right)^{1/n} + \xi(y)\frac{d}{dy}\left[\left(\frac{y}{1-y}\right)^{1/n}\right]$$

$$= \alpha[3y^2 - 2(1+r)y + r]\left(\frac{y}{1-y}\right)^{1/n}$$

$$+ [1 + \alpha y(y-1)(y-r)]\frac{1}{n(1-y)^2}\left(\frac{y}{1-y}\right)^{\frac{1-n}{n}}$$

The roots of this function on the interval $y \in [0, 1]$ were calculated numerically and used to determine the values of $x$ at the local extrema of the inverted S-shaped response. Either two extrema were found for which $x > 0$ (this was ensured by the choice of parameter $\alpha$, see S1 Text), in which case the width of the S-shaped response curve was determined by their difference, or no extrema were found, in which case the width equaled zero.

It should be noted that in case $x = [\text{CycB}]^*$ and $y = {[\text{Cdk1}]^*}/{d \cdot [\text{CycB}]^*}$, we calculate the derivative of $x$ with respect to $y$ while $y$ is expressed as a function of $x$. As shown in S1 Text, this has no consequences for finding the local extrema of $x$.

## Mass-action model for the PP2A-ENSA-GWL network

The model of the PP2A-ENSA-GWL network was largely based on the network described in [61], where the double negative feedback between GWL and PP2A can give rise to bistability. GWL indirectly inhibits PP2A by phosphorylating ENSA, which is both a substrate and inhibitor of PP2A and binds it in a complex C [62]. Here, the model was converted into an oscillator by incorporating synthesis and degradation of [Cdk1]:

$$\begin{cases} \dfrac{d[\text{Cdk1}]}{dt} = b_{\text{syn}} - b_{\text{deg}}[\text{Cdk1}][\text{APC}]^* \\[2mm] \dfrac{d[\text{GWLp}]}{dt} = k_{\text{pg}}[\text{GWL}][\text{Cdk1}] - k_{\text{dg}}[\text{GWLp}][\text{PP2A}] \\[2mm] \dfrac{d[\text{C}]}{dt} = k_{\text{ass}}[\text{ENSAp}][\text{PP2A}] - (k_{\text{dis}} + k_{\text{cat}})[\text{C}] \\[2mm] \dfrac{d[\text{ENSAp}]}{dt} = k_{\text{dis}}[\text{C}] + k_{\text{pe}}[\text{ENSA}][\text{GWLp}] - k_{\text{ass}}[\text{ENSAp}][\text{PP2A}] \\[2mm] \dfrac{d[\text{APC}]^*}{dt} = k_{\text{pa}}(1 - [\text{APC}]^*)[\text{Cdk1}] - k_{\text{da}}[\text{APC}]^*[\text{PP2A}] \end{cases} \quad (12)$$

with conservation of mass giving:

$$[\text{GWL}] = [\text{GWL}_{\text{tot}}] - [\text{GWLp}]$$

$$[\text{PP2A}] = [\text{PP2A}_{\text{tot}}] - [\text{C}]$$

$$[\text{ENSA}] = [\text{ENSA}_{\text{tot}}] - [\text{ENSAp}] - [\text{C}]$$

Parameter values were manually screened to obtain a steady-state response curve centered around $[\text{Cdk1}] \approx 20$ nM (to be in line with $K_{\text{cdk,apc}} = 20$ nM used elsewhere in this paper) and

**Table 1. Default parameter values for the mass-action model of the PP2A-ENSA-GWL network.**

| Parameter | Value | Units | Description |
|---|---|---|---|
| $k_{pg}$ | 0.07 | 1/(nM min) | Phosphorylation constant of GWL |
| $k_{dg}$ | 7.08 | 1/(nM min) | Dephosphorylation constant of GWL |
| $k_{pe}$ | 3.98 | 1/(nM min) | Phosphorylation constant of ENSA |
| $k_{pa}$ | 0.63 | 1/(nM min) | Phosphorylation constant of APC/C |
| $k_{da}$ | 1.58 | 1/(nM min) | Dehosphorylation constant of APC/C |
| $k_{ass}$ | 5.01 | 1/(nM min) | Association constant of ENSA and PP2A |
| $k_{dis}$ | 28.18 | 1/min | Dissociation constant of ENSA-PP2A complex C |
| $k_{cat}$ | 15.85 | 1/min | Catalytic constant of ENSA-PP2A complex C |
| $GWL_{tot}$ | 40 | nM | Total GWL concentration |
| $ENSA_{tot}$ | 200 | nM | Total ENSA concentration |
| $PP2A_{tot}$ | 40 | nM | Total PP2A concentration |

obtain oscillations with biologically relevant periods (Table 1). Concentration levels of $ENSA_{tot}$ are based on [108], and are five-fold higher than those of $PP2A_{tot}$ [62]. Steady-state response curves were determined via a custom Python script performing numerical continuation [109].

### Fitting the S-shaped module to the mass-action model

Once the steady-state response of the mass-action model was determined, we manually fitted a piecewise linear scaling function $\xi([APC]^*)$, that was subsequently incorporated into the non-dimensionalized system (i-b). Letting $x = [APC]^*$, and denoting the $[APC]^*$ value were $\xi$ reaches its local maximum $\xi_{max}$ as $x_{max}$ (similar for the minimum $x_{min}$) (see Fig 9C for a plot of the $\xi$ function), we have:

$$\xi = \begin{cases} \dfrac{\xi_{max} - 1}{x_{max}} x + 1 & \text{if } x \leq x_{max} \\[2em] \dfrac{\xi_{min} - \xi_{max}}{x_{min} - x_{max}} (x - x_{max}) + \xi_{max} & \text{if } x_{max} < x \leq x_{min} \\[2em] \dfrac{1 - \xi_{min}}{1 - x_{min}} (x - x_{min}) + \xi_{min} & \text{if } x_{min} < x \end{cases}$$

For Fig 4B, 4C and 4D, the best fit was obtained for $x_{min} = 0.60$, $x_{max} = 0.36$, $\xi_{min} = 0.75$, and $\xi_{max} = 1.28$. The location of the upper branch was manually shifted by multiplying the Hill-like term in system (i-b) (with $n = 5$) by a correction factor equaling 0.95. For the time traces, $\epsilon_{apc}^*$ was manually screened to give the best correspondence with the mass-action model. To reduce the separation of timescales in Fig 4G and 4H, the rate constants (except from synthesis and degradation) in Table 1 were divided by 16.5, a value as large as possible that still resulted in oscillations of the system. In Fig 4I, a divisor of 100 (rather than 16.5) was used.

## System equations for interlinked switches

The equations for the interlinked switches were derived following the same principles as described before, resulting in:

$$
\begin{cases}
\dfrac{d[\text{CycD}]}{dt} &= d_{\text{syn}} - d_{\text{deg}}[\text{CycD}]([\text{APC}]^* + \delta_d) \\[2ex]
\dfrac{d[\text{E2F}]^*}{dt} &= \dfrac{1}{\epsilon_{\text{e2f}}}\left( \dfrac{[\text{CycD}]^n}{\left(\xi([\text{E2F}]^*)\cdot K_{\text{cyc,e2f}}\right)^n + [\text{CycD}]^n} - [\text{E2F}]^* \right) \\[2ex]
\dfrac{d[\text{CycB}]}{dt} &= b_{\text{syn}}\cdot[\text{E2F}]^* - b_{\text{deg}}[\text{CycB}]([\text{APC}]^* + \delta_b) \\[2ex]
\dfrac{d[\text{Cdk1}]}{dt} &= \dfrac{1}{\epsilon_{\text{cdk}}}\left( \dfrac{[\text{CycB}]^n}{\left(\xi\!\left([\text{Cdk1}]/[\text{CycB}]\right)\cdot K_{\text{cyc,cdk}}\right)^n + [\text{CycB}]^n}[\text{CycB}] - [\text{Cdk1}] \right) \\[2ex]
\dfrac{d[\text{APC}]^*}{dt} &= \dfrac{1}{\epsilon_{\text{apc}}}\left( \dfrac{[\text{Cdk1}]^n}{\left(\xi([\text{APC}]^*)\cdot K_{\text{cdk,apc}}\right)^n + [\text{Cdk1}]^n} - [\text{APC}]^* \right)
\end{cases}
\tag{13}
$$

with

$$
\xi([\text{APC}]^*) = 1 + \alpha_{\text{apc}}[\text{APC}]^*([\text{APC}]^* - 1)([\text{APC}]^* - r)
$$

$$
\xi\!\left([\text{Cdk1}]/[\text{CycB}]\right) = 1 + \alpha_{\text{cdk}}\cdot\frac{[\text{Cdk1}]}{[\text{CycB}]}\left(\frac{[\text{Cdk1}]}{[\text{CycB}]} - 1\right)\left(\frac{[\text{Cdk1}]}{[\text{CycB}]} - r\right)
$$

$$
\xi([\text{E2F}]^*) = 1 + \alpha_{\text{e2f}}[\text{E2F}]^*([\text{E2F}]^* - 1)([\text{E2F}]^* - r)
$$

As before, $[\text{APC}]^* = [\text{APC/C}]/[\text{APC/C}]_{\text{tot}}$ and similarly $[\text{E2F}]^* = [\text{E2F}]/[\text{E2F}]_{\text{tot}}$. For all other variables and parameters, the original dimensions were retained. Note how we introduced a basal, $[\text{APC}]^*$-independent degradation of the cyclins here (i.e. $-d_{\text{deg}}\,\delta_d[\text{CycD}]$ and $-b_{\text{deg}}\,\delta_b[\text{CycB}]$), as some regulatory aspects of the cell cycle act upon these parameters (see below). Default parameter values were manually screened so that oscillations with biologically relevant periods were obtained (Table 2). As before, the thresholds $K_{\text{cyc,cdk}}$ and $K_{\text{cdk,apc}}$ were chosen in line with the experimental values reported in [11, 15]. The threshold value $K_{\text{cyc,e2f}}$ at which [CycD] activates $[\text{E2F}]^*$ was assumed to be 3 times higher than the threshold $K_{\text{cyc,cdk}}$ at which [CycB] activates [Cdk1], i.e. 120 nM, based on simulations from [56] where CycD levels are about three times higher than CycB levels and on simulations from [19] where CycD levels rise up to $\sim 100$ nM.

To distinguish the different cell cycle phases in this model, cells were considered to be in M phase wherever $[\text{APC}]^* > 0.95$, in G1 wherever $[\text{E2F}]^* < 0.95$ and $[\text{APC}]^* < 0.95$, and in S/G2 wherever $[\text{E2F}]^* > 0.95$ and $[\text{APC}]^* < 0.95$.

To model the effect of the restriction point, reduced levels of external growth factors were incorporated into the model by reducing the synthesis rate $d_{\text{syn}}$ of [CycD] by a factor 10. This number was chosen as the intermediate value of experimental measurements that demonstrated how growth factor stimulation increases CycD levels between 4 and 20 fold [110, 111]. DNA damage during G1 phase was accounted for by increasing the basal degradation rate of [CycD] (i.e. $\delta_d$) by a factor 3 and damage in G2 phase was modeled by increasing the width of the bistable [Cdk1] response (i.e. $\alpha_{\text{cdk}} = 30$) to shift the right fold of the curve to higher [CycB]

**Table 2. Default parameter values for modeling the cell cycle as interlinked bistable switches.**

| Parameter | Value | Units | Description |
|---|---|---|---|
| $r$ | 0.5 | | As defined in $\xi$ |
| $\alpha_{e2f}$ | 5 | | As defined in $\xi$ |
| $\alpha_{cdk}$ | 5 | | As defined in $\xi$ |
| $\alpha_{apc}$ | 5 | | As defined in $\xi$ |
| $n$ | 15 | | Hill exponent |
| $\epsilon_{e2f}$ | 0.01 | | Time constant E2F reaction |
| $\epsilon_{cdk}$ | 0.01 | | Time constant Cdk1 reaction |
| $\epsilon_{apc}$ | 0.01 | | Time constant APC/C reaction |
| $\delta_d$ | 0.05 | | Determining basal CycD degradation |
| $\delta_b$ | 0.05 | | Determining basal CycB degradation |
| $K_{cyc,e2f}$ | 120 | nM | Threshold of E2F activation by CycD |
| $K_{cyc,cdk}$ | 40 | nM | Threshold of Cdk1 activation by CycB |
| $K_{cdk,apc}$ | 20 | nM | Threshold of APC/C activation by Cdk1 |
| $d_{syn}$ | 0.15 | nM/min | Synthesis rate of CycD |
| $d_{deg}$ | 0.009 | 1/min | First order degradation of CycD |
| $b_{syn}$ | 0.03 | nM/min | Synthesis rate of CycB |
| $b_{deg}$ | 0.003 | 1/min | First order degradation of CycB |

levels. The value of $\alpha_{cdk} = 30$ was chosen so that the steady state of [CycB] in the standard model is below the [Cdk1] activation threshold.

## Simulations and analysis

All simulations were performed in Python 3.7. Ordinary differential equations were solved using the Python Scipy package solve_ivp (method = Radau). For equations including a delay, the JITCDDE package was used [112]. The amplitudes and periods were numerically determined from the local extrema in the time series via a custom Python script. Small amplitude oscillations and damped oscillations were omitted from the analysis.

## Coupling of the cell cycle to the circadian clock

Phase locking occurs when the ratio of periods (or frequencies) from two oscillators equals a rational number p:q, meaning that p cycles of one oscillator are completed while the second oscillator completes q cycles. To check for phase locking between the cell cycle and circadian clock, the system of interlinked switches (Eq 13) was reanalyzed (again with parameters from Table 2), but this time the prefactor $\alpha_{cdk}$ in $\xi(^{[Cdk1]}/_{[CycB]})$ was periodically shifted between its basal level and a predefined maximal level (i.e. $\alpha_{cdk} + 2A_{cdk}$) at a forcing frequency $\omega_{circadian}$ ranging from 1/3 to 3 times the natural frequency of the [Cdk1] oscillations:

$$\xi\left(^{[Cdk1]}/_{[CycB]}\right) = 1 + \alpha_{cdk}^* \frac{[Cdk1]}{[CycB]}\left(\frac{[Cdk1]}{[CycB]} - 1\right)\left(\frac{[Cdk1]}{[CycB]} - r\right)$$

with $\alpha_{cdk}^* = [\alpha_{cdk} + A_{cdk} + A_{cdk}\sin(\omega_{circadian}t)]$. Custom Python code was used to find a repeating pattern in the forced [Cdk1] oscillations, based on the difference between the simulated time series and a time shifted version of itself (similar to calculating the autocorrelation). The ratio of the periods from this repeating pattern and the circadian clock was compared with p:q ratios for p and q in {1,2,3,4,5} to decide whether phase locking occurred.

## Supporting information

**S1 Text. Supplementary information.** This file contains additional mathematical analysis of the models and the supplemental figures listed below.
(PDF)

**S1 Fig. Alternative definitions of the scaling function $\xi$. (A-D)** Except from the cubic definition of $\xi$ in the main text, alternative definitions can be used. An example of a quadratic function would be $\xi([APC]^*) = 1 + \alpha_{apc}([APC]^* - 1)([APC]^* - r)$. For a linear function, one possible definition would be $\xi([APC]^*) = \frac{1-\alpha_{apc}}{r-1}([APC]^* - 1) + \alpha_{apc}$. The equation for the piecewise linear approximation is given in the Methods section of the main text. In each case, $r = 0.5$. **(E-H)** Corresponding response curves obtained by multiplying a Hill function ($K = 20$, $n = 15$) with the scaling functions in panels A-D.
(TIFF)

**S2 Fig. Effect of parameters on the cubic scaling function and system response. (A-C)** Effect of parameter $r$ on the scaling function $\xi$ (A), S-shaped response curve in the phase plane (B) and oscillatory region in the parameter space (C). **(D-F)** Effect of the Hill coefficient $n$ on the original ultrasensitive response (D), the derived S-shaped response curve in the phase plane (E) and oscillatory region in the parameter space (F). **(G-I)** Effect of threshold $K$ on the original ultrasensitive response (G), S-shaped response curve in the phase plane (H) and oscillatory region in the parameter space (I).
(TIFF)

**S3 Fig. Oscillations for a time delayed, ultrasensitive cell cycle model. (A)** Block diagram of the ultrasensitive, delayed negative feedback network. **(B)** Oscillatory regions for different values of the Hill coefficient $n$. **(C)** Period of the [Cdk1] oscillations as a function of the relative synthesis rate $c$ and time delay. **(D,G)** [Cdk1] and [APC]$^*$ amplitudes as a function of the relative synthesis rate $c$ and time delay. **(E,F,H,I)** Time traces and phase planes for indicated parameter values.
(TIFF)

**S4 Fig. Oscillation period and amplitude for the S-shaped module. (A)** [Cdk1] period as a function of the relative synthesis $c$ and the width of the S-shaped region. **(B)** [Cdk1] amplitude as a function of the relative synthesis $c$ and the width of the S-shaped region. **(C)** [APC]$^*$ amplitude as a function of the relative synthesis $c$ and the width of the S-shaped region.
(TIFF)

**S5 Fig. Oscillation period and amplitude for the delayed S-shaped module. (A)** [Cdk1] period as a function of the width of the S-shaped region and delay. **(B)** [Cdk1] amplitude as a function of the width of the S-shaped region and delay. **(C)** [APC]$^*$ amplitude as a function of the width of the S-shaped region and delay.
(TIFF)

**S6 Fig. Irregular cell cycle oscillations in a chain of bistable switches.** In Fig 7 in the main text we indicated grey regions in parameter space for which irregular oscillations were observed. Here, we show time traces of [E2F]$^*$ for such irregular patterns.
(TIFF)

**S7 Fig. Effect of synthesis and degradation rates of CycD and CycB on the duration of different cell cycle phases.** In Fig 7 in the main text we showed the effect of changing synthesis and degradation rates on the overall length of the cell cycle. Here, we separate the effects on the different cell cycle phases. White areas represent regions where no oscillations can be

observed, while for the grey areas irregular oscillations exist.
(TIFF)

**S8 Fig. Extension of the cell cycle model by additional switches.** In Fig 7 in the main text we represented the cell cycle as a chain of three bistable switches. Here, we extended this model by including the hypothetical switch of FoxM1 activity with respect to CycA levels.
(TIFF)

**S9 Fig. Phase locking between the circadian clock and the cell cycle.** Time traces showing the absence or presence of p:q phase locking (with p and q in {1,2,3,4,5}) between the cell cycle and the circadian clock for several parameter combinations. Related to the Arnold tongues shown in Fig 8.
(TIFF)

**S1 Video. The cell cycle can be represented as a chain of interlinked bistable switches.** Video showing how the cell cycle progresses through the different bistable switches as shown in Fig 7.
(MP4)

## Acknowledgments

We are grateful to Prof. Catherine Verfaillie and the members of the Gelens lab for useful comments on the manuscript.

## Author Contributions

**Conceptualization:** Jan Rombouts, Lendert Gelens.

**Formal analysis:** Jolan De Boeck.

**Funding acquisition:** Jolan De Boeck, Jan Rombouts, Lendert Gelens.

**Investigation:** Jolan De Boeck, Jan Rombouts, Lendert Gelens.

**Methodology:** Jolan De Boeck, Jan Rombouts, Lendert Gelens.

**Project administration:** Lendert Gelens.

**Supervision:** Lendert Gelens.

**Visualization:** Jolan De Boeck, Jan Rombouts.

**Writing – original draft:** Jolan De Boeck.

**Writing – review & editing:** Jolan De Boeck, Jan Rombouts, Lendert Gelens.

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
