## [Decision Letter · Decision Letter 0]

25 May 2021

Dear Dr. Gelens,

Thank you very much for submitting your manuscript "A modular approach for modeling the cell cycle based on functional response curves" for consideration at PLOS Computational Biology.

As with all papers reviewed by the journal, your manuscript was reviewed by members of the editorial board and by several independent reviewers. In light of the reviews (below this email), we would like to invite the resubmission of a significantly-revised version that takes into account the reviewers' comments.

We cannot make any decision about publication until we have seen the revised manuscript and your response to the reviewers' comments. Your revised manuscript is also likely to be sent to reviewers for further evaluation.

Sincerely,

Attila Csikász-Nagy

Associate Editor

PLOS Computational Biology

Mark Alber

Deputy Editor

PLOS Computational Biology

Reviewer's Responses to Questions

**Comments to the Authors:**

Reviewer #1: In this work, a mathematical model of cell cycle regulation is developed by combining three functional modules (ultrasensitive, bistable and time delay). These modules are not derived from a detailed description of the underlying biochemical events, but provide a phenomenological description of a given desired behavior. It is shown that, by combining multiple switches and by including time delay, the proposed cell cycle model is able to describe events such as DNA damage and coupling to the circadian clock. In my opinion, the proposed strategy of combining multiple phenomenological modules to describe complex behaviors is promising and deserves investigation.

The major concerns about the reasoning and results are listed below, and concern the presentation and technical correctness of some derivations:

1 Abstract, Introduction (page 4, lines 113-117) and Results (page 5, line 122): even if the difference between an ultrasensitive and an S-shaped curve will be clarified later in the text, at this point, such a difference is, in my opinion, not clear. Indeed, an ultrasensitive response described by a sigmoidal function (e.g. the Hill function with Hill coefficient greater than 1) is itself an S-shaped curve.

2 Page 13 (system (iii)). As explicitly mentioned by the authors (see lines 334-335), the right hand side of the second ODE seems to not correspond to any of the modules (ultrasensitivity, S-shaped, delay) introduced in the first section, since the Hill-like equation is multiplied by [CycB]. Then, the model cannot be seen as a composition of the previously introduced modules. Furthermore, the choice of denoting the total concentration of CycB-Cdk1 complexes and the concentration of activated CycB-Cdk1 complexes by [CycB] and [Cdk1], respectively, (lines 330-331), is very confusing.

3 Pages 22-23. The ultrasensitive response [Cdk1]/[CycB] is not the same ultrasensitive curve as before (e.g., page 21, line 607 and below). The most remarkable difference is that, as noted by the authors (page 23, lines 632-633), with this expression, the Cdk1 curve does not saturate. On the contrary, it grows indefinitely. Then, its inverse is not exactly an S-shaped curve (I have generated a MATLAB plot to visualize this). Consequently, the plots of Figure 10 are not valid anymore and, indeed, in this case, the inverted ultrasensitive curve of Figure 10, panel (B) is not exactly an S-shaped curve. The reason for this behavior is mentioned by the authors at lines 628-629: “[CycB] is not a constant and therefore, the rate of change of [Cdk1]/[CycB] would need to be given by the quotient rule for differentiation.”

4 Page 24.It is clear how the expression of the non-dimensionalized inverted bistable response curve x has been computed. However, both the independent variable y and the dependent variable x are functions of [CycB], more precisely, y is a function of x. How is it possible to compute the derivative of x with respect to y given the fact that y a function of x?

5 Page 26, lines 671-672. “Default parameter values were manually screened so that oscillations with biologically relevant periods were obtained (Table 2).” It seems that the cyclins thresholds are the parameters that have been manually screened to generate oscillations. How are the thresholds of cyclins related to one another with respect to the biologically relevant periods? Are these thresholds compatible with the experimentally measured values?

6 Page 26, lines 680-682. The authors state that “This number was chosen based on experimental measurements that demonstrated how growth factor stimulation increases CycD levels between 4 and 20 fold [98, 99].” Could the authors be more precise with respect to the number used and to what it means in the context of testing the effect of the restriction point?

In addition, the following aspects require substantial explanations:

7 Page 4, lines 82-106. The difference between the two strategies is now clear: the first strategy consists in omitting some of the reactions, while the second strategy simplifies the model by introducing some assumptions (e.g. the steady state assumption). Can the use of phenomenological expressions be considered as a third strategy?

8 Page 11, (system (ii)).The role of delays τ_1 and τ_2 in the ODEs of (system (ii)) is not clear. Whilst their meaning is clearly explained in lines 295-300, connecting these explanations to the way in which τ_1and τ_2 appear in the ODEs is not straightforward. In addition, when τ_1 and τ_2 assume different values, the time delay in both ODEs becomes time-varying (since τ is a function of [APC]*); is this biologically reasonable?

9 Page 14, Figure 7. Comprehension and interpretation of the plots seems to be rather hard. Moreover, panel (B) refers to both system (iii-b) and (iii-c): does this mean that the two systems exhibit equal [Cdk1] amplitude?

10 Page 15. Several concerns arise regarding systems (iii-a), (iii-b) and (iii-c). First of all, at line 340, it should be clarified that the label “system (iii-a)” refers to Figure 7A. At lines 341-344, the parameter d has been introduced; it seems to be confusing to define the relative synthesis first as c and then as the ratio between c and d? Also, the considerations at lines 351-357 are not clear.

11 Pages 17 and 19 ODEs are missing for the model in which the circadian regulation of Wee1 has been introduced (see lines 460-461). Also, the concept of Arnold tongues (lines 473-490) would require additional explanations. Finally, the expression “the strength of the coupling (here Acdk, i.e. the extent to which the [Cdk1] activation threshold is shifted to higher [CycB] levels over one period)” (line 471-472).

12 Page 19, line 493. “Under the premise that the functionalities of these individual modules do not change when combined with each other”. Unfortunately, this assumption is not always verified, as explained few lines below (see e.g. 499-500).

13 Page 26. in the system equations for interlinked switches, what is the meaning of parameters δ_d and δ_b? It would be better to introduce the meaning of these parameters within the text, in addition to Table 2. These terms were present neither in (systems (ii)) nor in (system (iii)). Why were they added in the interlinked switches system? Furthermore, the right hand side of the fourth ODE does not correspond to any of the modules (ultrasensitivity, bistability, delay) introduced in the first section since, again, the Hill-like equation is multiplied by [CycB] (see the previous comment regarding (system (iii))).

14 Page 6, lines 136-138. How have the curves in Figure 2B been obtained? I guess the authors have computed the inverse of the function in Eq. (1) (namely, Input as a function of Output) and then they have plotted this latter inverse function for different values of α. I would suggest to explain this explicitly.

15 Page 10, section “The S-shaped module reproduces the behavior of mass-action models”: In this section, the authors claim (lines 269-270 and lines 289-290) to show that the modified Hill function (namely, the "phenomenological" expression in Eq (2), which provides an S-shaped response) and its piecewise linear approximation are able to reproduce the oscillatory behavior of the mass-action model for the PP2A-ENSA-GWL network. Is there any reason why the S-shaped response should be preferred to the mass-action model?

16 Page 25, Table 1. The value of ENSA_tot used by the authors is equal to 40nM, while in reference [61] this value is equal to 1000nM. As the authors refer to [61], which is then the specific reason to change the value of ENSA_tot compared to that paper?

Finally, the following minor comments require rephrasing, adjustments and/or a better explanation of concepts:

17 Page 7, lines 166-167. “the rate of change of [APC]* is proportional to the difference between the current [APC]* level and its experimentally determined ultrasensitive steady state response”. Does this mean that the term - [APC]* is not representing degradation?

18 Page 9, line 235.What does "extrema of ξ([APC]^*)” mean?

19 Page 11, lines 308-309. “Without a time delay, i.e. τ=0, system (ii) is identical to system (i)”. This claim is trivial.

20 Page 15, line 366. “and” should be replaced with “as”.

21 Page 16: the concept of “oscillations orbiting around the response curves/bistable switches” is not clear; please, see the caption of Figure 8 (“Grey regions represent oscillations that are not orbiting around all three bistable switches.”) and also lines 402-403 and lines 405-407

22 Page 23, line 635. What does “apparent” mean?

23 Page 27, line 696. Can the authors please indicate what do they mean with “shifted”?

24 Supplemental Information, page 1, few lines below equation (1). “we will show that the steady state is linearly stable”. Even if stability of the steady state is assessed through linearization, the expression “linearly stable” seems to be not correct, since model (1) is nonlinear.

25 Page 4, line 108: please, replace “cycles” with “cycle” (singular).

26 Please replace “a S-shaped response/module” with “an S-shaped response/module”: page 6, line 131; page 8, line 184 and line 186; page 11, line 308; page 15, line 385 and line 387; page 21, line 603; page 22, line 614;

27 Page 7, caption of Figure 3(E): please, replace “bistable [APC]* nullclines” with “S-shaped [APC]* nullcines”.

28 Page 12, line 326: “a S-shaped response”. Maybe, “bistable” would be more appropriate in this case.

29 Page 13, Figure 6(B): please, replace “bistable width” with “width of the S-shaped region”.

30 Page 15, line 348: please, replace “bistable” with “S-shaped”.

31 Page 15, lines 385-386: “can be depicted as a S-shaped module”. I would suggest to rephrase as "can be depicted as an S-shaped (in this case, also bistable) module".

32 Page 12, equation (12): in the third and the fourth differential equation, the parameter k_ass is present, whose value is not indicated in Table 1. Also, in the fifth differential equation, the parameter k_da appears, whose value is not present in Table 1.

Reviewer #2: See attachment

Reviewer #3: Attached

**Have the authors made all data and (if applicable) computational code underlying the findings in their manuscript fully available?**

Reviewer #1: **No: **The computational code is not available.

Reviewer #2: **No: **All models should be provided as machine-readable Python programs in Supplementary Files

Reviewer #3: Yes

PLOS authors have the option to publish the peer review history of their article (what does this mean?). If published, this will include your full peer review and any attached files.

Reviewer #1: No

Reviewer #2: No

Reviewer #3: No
---

## [Decision Letter · Decision Letter 1]

8 Jul 2021

Dear Dr. Gelens,

Thank you very much for submitting your manuscript "A modular approach for modeling the cell cycle based on functional response curves" for consideration at PLOS Computational Biology. As with all papers reviewed by the journal, your manuscript was reviewed by members of the editorial board and by several independent reviewers. The reviewers appreciated the attention to an important topic. Based on the reviews, we are likely to accept this manuscript for publication, providing that you modify the manuscript according to the review recommendations.

Please specifically focus on:

- mentioning the alternatives of a limit cycle representation of the cell cycle

- providing the codes of the models, either as supplement or on a repository

- answering the three major comments of .referee 1

Sincerely,

Attila Csikász-Nagy

Associate Editor

PLOS Computational Biology

Mark Alber

Deputy Editor

PLOS Computational Biology

[LINK]

Reviewer's Responses to Questions

**Comments to the Authors:**

Reviewer #1: I have carefully read the revised manuscript, and I thank the authors for having clarified some aspects that I have raised in the first round of revision. A few explanations have been now added where required, and almost all issues raised in the previous round of review have been carefully addressed.

However, despite improvements have been made, a number of technical concerns remains, mainly regarding the S-shaped module and the way in which time delays have been incorporated into the model. Specifically:

- Regarding (system (iii)) – former major concerns 2) and 3) – , I believe that confusion arising from the names of the state variables has been resolved, and I thank the authors for having clarified this point. Concerning the ODEs, I understand the reason for multiplying the RHS of the ODE by [CycB]; the explanation provided by the authors is reasonable. However, according to this reviewer, it seems to be incorrect to describe (system (iii)) as the composition of the previously introduced modules, since the RHS of the ODE describing the dynamics of [CycB] does not correspond to any of these modules.

- Regarding the computation of the derivative of the expression for the non-dimensionalized inverted bistable response curve – former major comment 4) –, I thank the authors for having inserted a detailed technical explanation (page 29, lines 941-943). However, according to this reviewer, a technical mistake exists in their reasoning. The claim “x is a function of y, and y a function of z” is only partly correct, since y is a fucntion of both x and z (i.e. [CycB] and [Cdk1], respectively). If x is a function of y and y is a function of z, then the application of the chain rule of differentiation yields, as correctly indicated by the authors, dx/dz = (dx/dy)(dy/dz). However, this seems to be not the case the authors are dealing with. The problem in their formulation (which is absolutely correct when x=[Cdk1] and y=[APC]) is that x=[CycB] is treated as an independent variable in the RHS of the equation whereas it is treated as the dependent variable in the LHS of the equation.

- Regarding the section “Delay increases the period, amplitude and robustness of oscillations” – former substantial explanation 8) –, I acknowledge that it has been reorganized in a clearer way. However, concerns still remain regarding the temporal delays τ_1 and τ_2. Specifically, according to this reviewer, the way time delays have been incorporated into the model does not reflect the provided biological explanation. Since the temporal delay τ_1 is the delay on the “left branch” of the network in Figure 5A (i.e., form Cdk1-CycB to APC) and τ_2 is the delay on the “right branch” of the network in Figure 5A (i.e., form APC to Cdk1-CycB), I would have expected τ_1 in the second ODE and τ_2 in the first ODE. In the equations of (system (ii)), the dynamics of APC/C is alternatively affected by the “activation delay” and by the “inactivation delay”. In this way, τ_2 does not reflect the meaning of this delay, which is the delay in the inactivation of CycB-Cdk1 by APC/C. Furthermore, it seems to be reasonable that, if the time delay depends on the state of the system, then this dependence exists both in case τ_1 equals τ_2 and in case τ_1 differs from τ_2. On the contrary, in the provided expression for the time delay τ, the time delay depends on the state of the system only if τ_1 and τ_2 are different. For which biological reason is this value independent of the state of the system only if the “activation delay” and the “inactivation delay” have the same value?

In addition, I still have the following minor comments:

1) Regarding the difference between an ultrasensitive and an S-shaped response curve – former major concern 1) –, I understand the point made by the authors when claiming that “an ultrasensitive curve is not really S-shaped”. However, I believe that this distinction is not universally recognized as, in some cases, the terms “ultrasensitive” and “S-shaped” are considered synonyms (see, e.g., [Perelson SciTranslMed 2011], [Gunawardena PNAS 2005], [Estrada Cell 2016]). For the sake of clarity - and to avoid any possible confusion -, I would suggest to explicitly mention that by “S-shaped curve” you mean a curve “with two bumps” (or, alternatively, a more appealing description), which is hence different from an ultrasensitive curve.

2) Page 13, Figure 5B (Figure 6B in the previous version of the manuscript): please, replace “bistable width” with “width of the S-shaped region” in the label of the X-axis.

3) Page 13, caption of Figure 4: please, indicate the value of the constant factor by which all mass-action constants were divided.

4) Page 14, lines 422-433: Please, rephrase the sentence “Such delays can be modeled in a phenomenological way by delay differential equations” with, e.g. The effects of these delays can be modeled in a phenomenological way by using delay differential equations”.

5) Page 19, line 576: Regarding Figure 7B, please add also the time course of APC/C, which is currently missing.

6) Page 20, line 606: please, replace “Fig. S8 Fig” with “Fig. S8”.

7) Page 22, line 694: If I correctly understood, “frequencies” should be replaced with “ratio between the two frequencies”.

8) Page 23, line 751: please, replace “numbers” (plural) with “number” (singular).

9) Page 24, line 784: please, replace “reviewed in this journal [63]” with “reviewed in [63]”.

10) Page 24, line 817: “are in line with previous work”. Could the authors provide a few references?

11) Page 31, lines 984-985 – former previous major concern 5) – Related to the sentence “Default parameter values were manually screened so that oscillations with biologically relevant periods were obtained (Table 2), please indicate the experimental references [11] and [15] that refer to this point, as you have specifically indicated these in the response letter.

Reviewer #2: Congratulations on an interesting and thought-provoking paper. I still disagree with you that the mammalian cell cycle can be modeled as a limit cycle oscillator (which also casts suspicion on CC-CR coupling simulations), but I approve publication nonetheless because I think your modular approach deserves to be studied and assessed by the computational biology community.

In the section on "S-shaped, but not ultrasensitive, responses cause a two-dimensional system of the cell cycle to oscillate", you might consider referring to Novak & Tyson (Nat Rev Mol Cell Biol, 2008) where the same point is made in a more general context.

Reviewer #3: The authors addresess all my comments.

**Have the authors made all data and (if applicable) computational code underlying the findings in their manuscript fully available?**

Reviewer #1: **No: **The computational code is not available.

Reviewer #2: Yes

Reviewer #3: None

PLOS authors have the option to publish the peer review history of their article (what does this mean?). If published, this will include your full peer review and any attached files.

Reviewer #1: No

Reviewer #2: **Yes: **John J. Tyson

Reviewer #3: No

Figure Files:

Data Requirements:

Reproducibility:

References:

---

## [Editor Report · Decision Letter 2]

19 Jul 2021

Dear Dr. Gelens,

We are pleased to inform you that your manuscript 'A modular approach for modeling the cell cycle based on functional response curves' has been provisionally accepted for publication in PLOS Computational Biology.

Best regards,

Attila Csikász-Nagy

Associate Editor

PLOS Computational Biology

Mark Alber

Deputy Editor

PLOS Computational Biology

---

## [Editor Report · Acceptance letter]

6 Aug 2021

PCOMPBIOL-D-21-00681R2 

A modular approach for modeling the cell cycle based on functional response curves

Dear Dr Gelens,

I am pleased to inform you that your manuscript has been formally accepted for publication in PLOS Computational Biology. Your manuscript is now with our production department and you will be notified of the publication date in due course.

With kind regards,

Andrea Szabo
